# A Building-Block Urban Meteorological Observation Experiment (BBMEX) Campaign in Central Commercial Area in Seoul

**Moon-Soo Park [1],\*** , **Jae-Young Byon [2]** , **Baek-Jo Kim [3]** , **Woosuk Choi [4]** , **Kwang-Min Myung [5]** , **Sang-Hyun Lee [6]** , **Tae-Il Cho [7]** , **Jung-Hoon Chae [1]** , **Jae-Sik Min [1]** , **Minsoo Kang [1]** , **Joon-Bum Jee [1]** , **Sang-Heon Kim [1]** and **Chang-Rae Cho [1]**

1. Research Center for Atmospheric Environment, Hankuk University of Foreign Studies, Yongin 17035, Korea; zhzhah79@gmail.com (J.-H.C.); min_jaesik@hufs.ac.kr (J.-S.M.); kangms8993@gmail.com (M.K.); rokmcjjb717@gmail.com (J.-B.J.); paulksh@hufs.ac.kr (S.-H.K.); crcho@hufs.ac.kr (C.-R.C.)
2. Applied Meteorological Research Division, National Institute of Meteorological Sciences, Jeju 63568, Korea; byonjy@kma.go.kr
3. High Impact Weather Research Center, National Institute of Meteorological Sciences, Gangneung 25457, Korea; bjkim@kma.go.kr
4. Seoul Institute of Technology, Seoul 01811, Korea; wschoi@sit.re.kr
5. Environment Business Dep., kt, Seoul 01811, Korea; bennym@hanmail.net
6. Department of Atmospheric Science, Kongju National University, Gongju 314701, Korea; sanghyun@kongju.ac.kr
7. Observer, Seoul 01811, Korea; parkkimbest@gmail.com
\* Correspondence: ngeograph2@gmail.com or moonsoo@hufs.ac.kr; Tel.: +82-31-8020-5589

**Abstract:** High-resolution meteorological information is essential for attaining sustainable and resilient cities. To elucidate high-resolution features of surface and air temperatures in high-rise building blocks (BBs), a 3-dimensional BB meteorological observation experiment (BBMEX) campaign was designed. The campaign was carried out in a central commercial area in Seoul during a heat-wave event period (5–6 August) in 2019. Several types of fixed instrument were deployed, a mobile meteorological observation cart (MOCA) and a vehicle were operated periodically. The surface temperature was determined to be strongly dependent on the facial direction of a building, and sunlit or shade by surrounding obstacles. Considerable increases in surface temperature on the eastern facades of buildings before noon, on horizontal surfaces near noon, and on the western facades in the afternoon could provide more energy in BBs than over a flat surface. The air temperatures in the BB were higher than those at the Seoul station by 0.1–2.2 °C (1.1–1.9 °C) in daytime (night-time). The MOCA revealed that the surface and air temperatures in a BB could be affected by many complex factors, such as the structure of the BBs, shades, as well as the existence of facilities that mitigate heat stresses, such as ground fountains and waterways.

**Keywords:** BBMEX; building-block meteorology; heat wave; surface temperature; thermal infrared imager; tropical night

## 1. Introduction

The urban population exceeded 4.2 billion in 2019 and is projected to reach 6.3 billion (65% of world population) by 2050 [1,2]. Meteorological features in urban areas show a very complex pattern in comparison with those in rural areas due to high-rise buildings, anthropogenic heat storage and release, and impervious surfaces e.g., [3,4]. The urban heat island (UHI) phenomenon, where the temperatures

in urban areas are higher than those in rural areas, is a well-known and prominent feature [5–8]. Higher temperatures in urban areas and relatively lower temperatures in rural areas induce urban–rural circulation over highly populated urban areas e.g., [9].

Most UHI studies assume that urban and rural areas are composed of land cover with kilometer-scale horizontal homogeneities to explain the temperature difference between two areas [10,11]. In real scenarios, urban areas are composed of various land-cover blocks: compact high-rise, open high-rise, compact low-rise, open low-rise buildings, commercial or industrial blocks, urban parks, water bodies, and so on [3]. The height and density of buildings, and land cover types become the main criteria for classifying urban climate zones. These are key factors in determining the surface roughness and zero-plane displacement lengths for describing wind and sky views, as well as temperature inside or over building-blocks (BBs) [12].

Recently, high-resolution meteorological information has become essential for attaining sustainable and resilient cities [13,14]. To support the high-resolution meteorological information service in urban areas, many observation experiments have assembled the horizontal distribution and vertical profiles of meteorological variables in urban areas since the 1970s (Table 1). Most urban meteorological observation networks, such as the Helsinki testbed in Helsinki, Shanghai Urban Integrated Meteorological Observation Network (SUIMON) in Shanghai, Study of Urban Impact on Rainfall and Fog/Haze (SURF) in Beijing, Tokyo Metropolitan Area Convection Study (TOMACS) in Tokyo, and High-resolution Urban Meteorological Observation System Network in the Seoul Metropolitan Area (UMS-Seoul) in Seoul, cover meso-$\gamma$ (20 to 2 km) or micro-$\alpha$ (2 km to 200 m) scales [13,15–17].

High-rise building blocks are problematic due to their meter-scale inhomogeneity both horizontally as well as vertically. Advances from micro-$\alpha$ scale to micro-$\beta$ or micro-$\gamma$ scales (200 m to 2 m) or higher are needed. There were several micro-scale observation projects in urban areas [18]. As examples, the Lausanne Urban Canopy Experiment (LUCE) had installed 92 stations over 300 m × 400 m areas from October 2006 to April 2007 to determine the sensible heat flux according to the surface geometry and type in Switzerland [19]. And the USscan network had installed 200 sensors over 250 m × 430 m area from July to August 2007 to investigate the fine-gridded temperature pattern in Tokyo downtown [20]. Recently, 52 stations in a block with 500 m × 700 m in Sydney had been installed during two sunny days in summer to verify the precinct ventilation performance [21]. The above datasets still had a spatial resolution of 10-meter scales.

For the purpose of advancing to a meter-scale meteorology, more observation datasets are still needed. These datasets could be used to quantify the air temperature difference between several points in a BB and to the nearest synoptic weather station from the BB, to quantify the effects of materials and structures of BB on surface and air temperatures in the BB, and to quantify the effects of facilities mitigating heat stresses on the temperatures in the BB.

To acquire surface and air temperatures with high temporal and meter-scale spatial resolution in a high-rise BB, a building-block urban meteorological observation experiment (BBMEX) campaign was designed. The 2019 BBMEX campaign was carried out over the central commercial area in Seoul City during a heat wave and tropical night periods in 2019. The campaign was co-organized by the National Institute of Meteorological Sciences and Hankuk University of Foreign Studies. The Seoul Institute of Technology (SIT), kt (telecommunication company), Kongju National University, and Observer Inc. (instrument company) participated in the campaign.

This study aims to introduce experimental outlines of the 2019 BBMEX campaign. The obtained surface and air temperature dataset is to be overviewed with a view to the structure of BB, facilities mitigating heat stresses, and environmental factors. The usefulness of the collected dataset for improving micro-$\gamma$ scale meteorology, and applicability to urban planning for attaining sustainable and resilient cities are demonstrated.

**Table 1.** Summary of urban meteorological observation experiments.

| Name | Major Instruments | Location | Period | Resolu-tion | Reference |
|---|---|---|---|---|---|
| METROMEX[1] | Rain gauge Meteorology | St. Louis, USA | Summer 1971–1975 | | Changnon (1981) [22] |
| BUBBLE[2] | Meteorology Energy balance Wind profiler Lidar | Basel, Switzerland | Summer 2001–summer 2002 | ~0.5 km | Rotach et al. (2005) [23] |
| URBAN2000 VTMX[3] | Meteorology Energy balance Trace sampler SODAR | Salt Lake City, USA | October 2000 | ~0.2 km | Allwine et al. (2002) [24], Doran et al. (2002) [25] |
| Helsinki testbed | Meteorology Radar Vertical profiles | Helsinki, Finland | 2007- | ~1.2 km | Koshikinen et al. (2011) [26] |
| TOMACS[4] | Meteorology Radar | Tokyo, Japan | 2010s | | Nakatani et al. (2015) [27], Misumi et al. (2019) [17] |
| SUIMON[5] | Meteorology Energy balance Vertical profile | Shanghai, China | | | Tan et al. (2015) [15] |
| UMS-Seoul[6] | Meteorology Energy balance Vertical profile Mobile | Seoul, Korea | 2014- | ~1.5 km | Park et al. (2017) [13] |
| LUCE[7] | Meteorology | Lausanne, Switzerland | October 2006 –April 2007 | ~40 m | Nadeau et al. (2009) [19] |
| USscan | Meteorology | Tokyo, Japan | July–August 2007 | ~20 m | Thepvilojanapong et al. (2010) [20] |
| PVP[8] | Meteorology | Sydney, Australia | February 2019 | ~80 m | He et al. (2019) [21] |

[1] METROMEX: Metropolitan Meteorological Experiment; [2] BUBBLE: Basel UrBan Boundary Layer Experiment; [3] VTMX: Vertical Transport and Mixing; [4] TOMACS: Tokyo Metropolitan Area Convection Study for Extreme Weather Resilient Cities; [5] SUIMON: Shanghai Urban Integrated Meteorological Observation Network; [6] UMS-Seoul: High-resolution Urban Meteorological Observation System Network in the Seoul Metropolitan Area; [7] LUCE: Lausanne Urban Canopy Experiment, [8] PVP: Precinct Ventilation Performance.

## 2. Experiment Design

### 2.1. Domain: Gwanghwamun Square and Sejong-Daero Streets

The Gwanghwamun (GHM) region was chosen for the 2019 BBMEX campaign (Figure 1) because it is located in the heart of central Seoul City in Korea and is surrounded by many high-rise BBs. GHM Square, with a south-north directional length of 550 m and west-east directional width of 33 m, is located in the middle of the region. The topographic height of the GHM ranges from 30 m at the southern edge to 33 m at the northern edge (http://map.vworld.kr). Most surfaces are covered with flat rectangular-type stone blocks. Short grasses (flower carpet) are planted over a 17 m (E–W) × 160 m (S–N) area in the northern part of the square (Figure 1c). A cooling fog and a ground fountain (the 12.23 Fountain) are facilitated in the southern part to mitigate heat stresses during daytime in hot summer (Figure 2b,c). Two statues are located at the center and southern part. Two historic waterways with a width, depth and length of 1 m, 2 cm, and 365 m, respectively, flow along the eastern and western border of the square (Figure 2d). Two Sejong-daero Streets with a width of 18 m surround the square (Figure 1c). The roads are covered with asphalt. A sidewalk with a width of 15 m surrounds each street. Sidewalks are mostly covered with stone blocks.

The Seoul automatic synoptic observation system (ASOS) station is located 0.95 km west of the square (Figure 1b). This station produces the Korea Meteorological Administration (KMA) official meteorological data of Seoul. It is surrounded by residential area except for green belt to the east.

The meteorological tower is installed over flat surface covered with short grasses. The topographical elevation of the Seoul ASOS Station is 85.7 m, which is higher than that of the square by approximately 53 m. Seoul City Hall is located 420 m south of the southern edge of the square (Figure 1b).

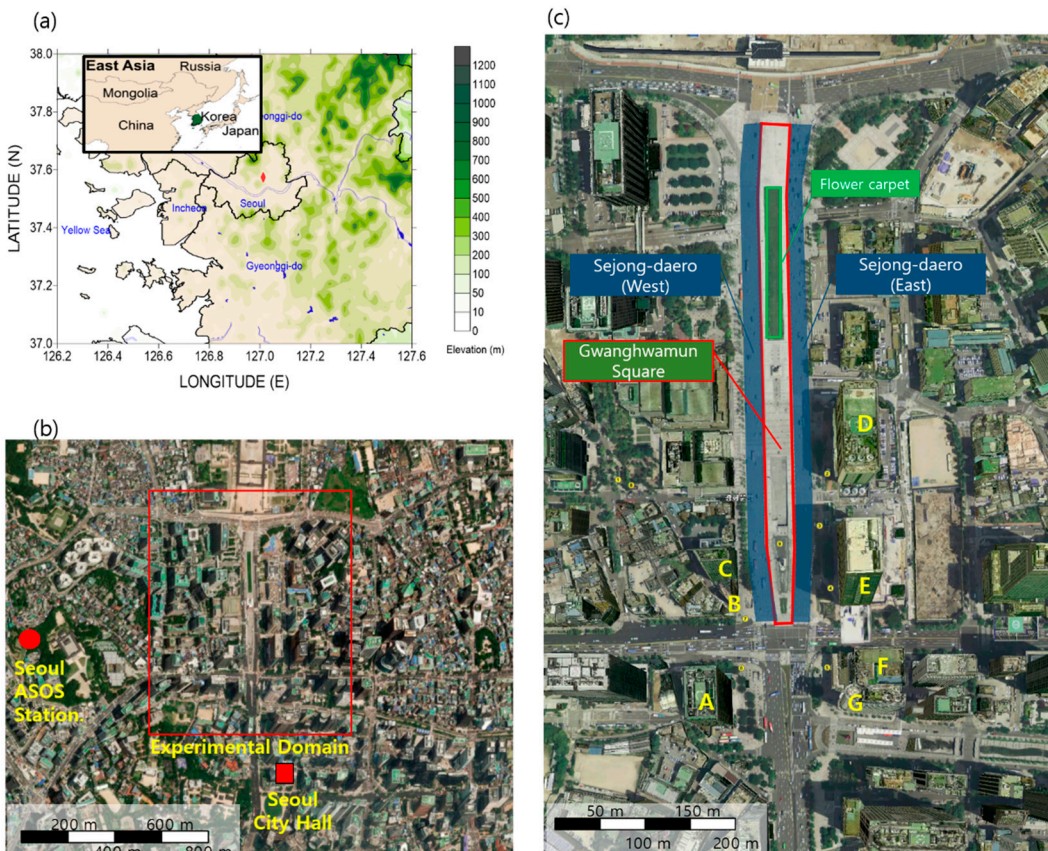

**Figure 1.** (**a**) Location of experimental site (red closed diamond) in Seoul Metropolitan Area, Korea, East Asia; (**b**) satellite image near the experimental domain; (**c**) location of Gwanghwamun Square and Sejong-daero Streets (East and West) and Buildings A–G in the enhanced 3-dimensional building map near the experimental site.

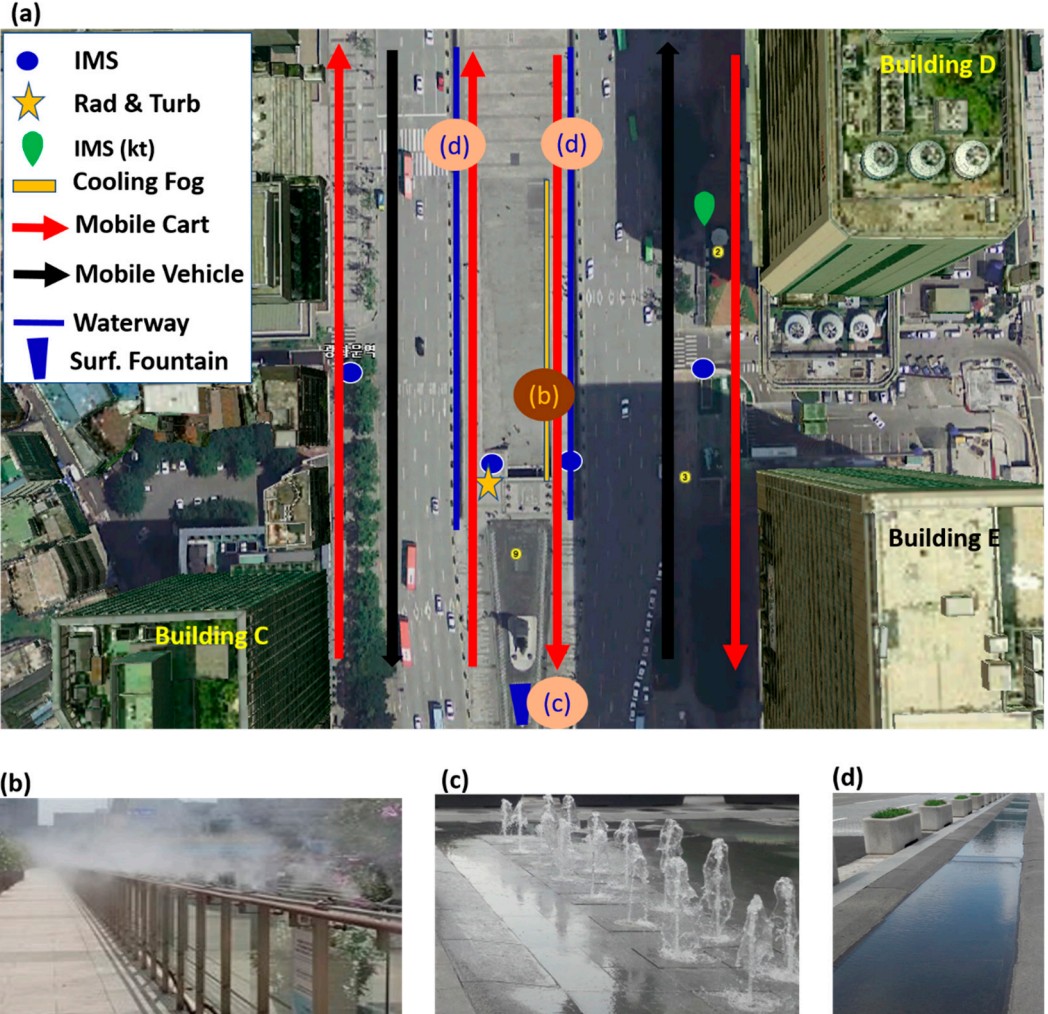

**Figure 2.** (**a**) Detailed location of fixed instruments or sensors and moving routes of mobile vehicle and cart with cooling fog measurement system, historic waterway, and surface fountain during the building-block urban meteorological observation experiment (BBMEX), (**b**) cooling fog, (**c**) surface fountain, and (**d**) historic waterway.

## 2.2. Instrumentations

To acquire a 3-dimensional meteorological feature in a BB, 3 types of meteorological observation system were installed and assembled (Table 2): the GHM station, 7 fixed automatic weather stations, and 2 mobile meteorological observation systems. GHM station is a part of the UMS-Seoul network [13]; it is installed on the rooftop of Building D, whose height is 70 m (Figure 1c). The GHM station has a surface energy balance system, ceilometer, wind lidar, and microwave radiometer [13]. The surface energy balance system includes a thermal infrared Imager (TIR), real-image camera, 3-dimensional sonic anemometer with a $CO_2/H_2O$ gas analyzer, temperature and relative humidity sensor, wind speed and direction sensor. The TIR (model: TS9230, manufacturer: Nippon Avionics) detects spectral energy within 320 (horizontal) × 240 (vertical) pixels with a spectral range between 8 and 13 μm (Figure 3d). The received energy by the TIR camera is a sum of the object, ambient, and atmospheric contributions [28]. The TIR radiation can be reduced to object-only contributions by neglecting the absorptivity of the atmosphere when the distance between the sensor and the object is sufficiently distant. The surface temperature can be retrieved with the assumption of unit emissivity. A retrieved surface temperature image for the field of view and a real camera image with the same field of view are saved every 10 min.

**Table 2.** Specification of main instruments system.

| System | Sensor or Specification | Reference Sampling Rate |
|---|---|---|
| Surface energy balance system (Gwanghwamun (GHM) Station) | Thermal Infrared Imager (TS9230, Nippon Avionics), 3-D sonic anemometer and $CO_2$/$H_2O$ Gas Analyzer (CSAT3A/EC150, Campbell Sci.), temperature and relative humidity (HMP155A, Campbell Sci.), wind speed and direction (W300P/A100M, Vector Instruments), air pressure (CS105, Campbell Sci.), precipitation (WDSA-205, Wedaen) | Park et al. (2017) [13] |
| Thermal Infrared Imager (TS9230, Nippon Avionics) | Spectral range: 8−13 µm Field of view: 21.7 ° (h) × 16.4 ° (v) Pixels: 320 (h) × 240 (v) Temperature range: −40 to 500 °C Temperature accuracy: ±2 °C | 10 min |
| Radiation and Sonic Anemometer (RS) | 3-dimensional sonic anemometer (model CSAT3A, Campbell scientific), net radiometer (downward/upward, shortwave/longwave) (model CNR4, Kipp and Zonen) | 10 Hz (sonic), 1 min (radiation) |
| Automatic Weather Station (Type A) | Temperature, relative humidity, wind speed, wind direction (model WXT536, Vaisala; temperature accuracy: ±0.3 °C) | 1 min |
| Automatic Weather Station (Type B) | Temperature (MWS: Pt100 sensor, accuracy: ±0.3 °C for −40 to 60 °C; ktWS: silicon bandgap sensor, accuracy: ±0.3 °C for 20 to 40 °C, ±1.0 °C for 0 to 70 °C), relative humidity, $PM_{10}$/$PM_{2.5}$ concentration, noise level | 1 min |
| Mobile Meteorological Observation Cart (MOCA) | Surface temperature (model SI-111-SS, Apogee; accuracy: ±0.2 °C for −10 to 65 °C), air temperature (0.5 m, 1.5 m, 2.5 m) (Pt100, accuracy: ±0.3 °C for −40 to 60 °C), Global Positioning System (GPS) | 1 s |
| Mobile Observation Vehicle (MOVE) | Temperature, Relative Humidity, Air pressure, precipitation, solar radiation (downward/upward shortwave/longwave), insolation, Global Navigation Satellite System (GNSS), GPS, road surface sensor (temperature, status, salinity, water depth, conductivity) | Kim et al. (2020) [29] 1 s |

A radiation and sonic anemometer system (RS) was used with a net radiation sensor at a height of 0.7 m, a 3-dimensional sonic anemometer with a height of 1.2 m, and an integrated meteorological sensor at 2.0 m high (Figure 3e). A net radiation sensor (model: CNR4, manufacturer: Kipp and Zonen) observes downward shortwave (0.3−2.8 µm) $S_\downarrow$, upward shortwave $S_\uparrow$, downward longwave (4.5−42 µm) $L_\downarrow$, and upward longwave $L_\uparrow$ radiative flux. A net radiative flux $R_{NET}$ is calculated by:

$$R_{NET} = S_\downarrow - S_\uparrow + L_\downarrow - L_\uparrow. \tag{1}$$

The albedo is calculated by the ratio of upward shortwave radiation to downward shortwave radiation when the downward shortwave radiation is larger than 50 W m$^{-2}$. A 3-dimensional sonic anemometer (model: CSAT3A, manufacturer: Campbell Sci.) observes $u$-, $v$-, $w$- components of winds and sonic temperature $T_S$ at 10 Hz. The friction velocity $u_*$, friction temperature $T_*$, and sensible heat flux $H$ are calculated using the equation:

$$u_* = \left[ (-\overline{w\prime u\prime})^2 + (-\overline{w\prime v\prime})^2 \right]^{1/4}, \tag{2}$$

$$T_* = \overline{w\prime T_S\prime}/u_*, \tag{3}$$

$$H = \rho C_p \overline{w\prime T_S\prime}, \tag{4}$$

where $\rho$ is the air density, $C_p$ is the specific heat at a constant pressure, and ( )$\prime$ indicates a deviation in a variable from a 30 min block average, and $\overline{a\prime b\prime}$ is the covariance between the two variables $a$ and $b$ [30].

Each automatic weather system (AWS) includes an integrated meteorological observation sensor (model: WXT536, manufacturer: Vaisala; model: MWS, manufacturer: Observer; model: ktWS, manufacturer: kt) (Figure 3f). Type A sensors observe the temperature, relative humidity, air pressure, wind speed and direction, and precipitation (WXT536). Type B sensors observe the air temperature, relative humidity, and air pressure (MWS and ktWS). Type A sensors were deployed at A1, A2, A3, and A4. Type B MWS sensors were deployed at heights of 0.5 m and 1.5 m at G1 (above grass) and G2 (above pavement) (Figure 3a). A Type B ktWS sensor was deployed at T1 (Figures 2a and 3a). In addition, 7 ktWS sensors were deployed at 7 different heights (13, 22, 36, 45, 54, 63, and 70 m) outside Building D to acquire the temperature profiles in the street canyon (Figure 3h). The highest sensor (70 m) was deployed over the rooftop of Building D.

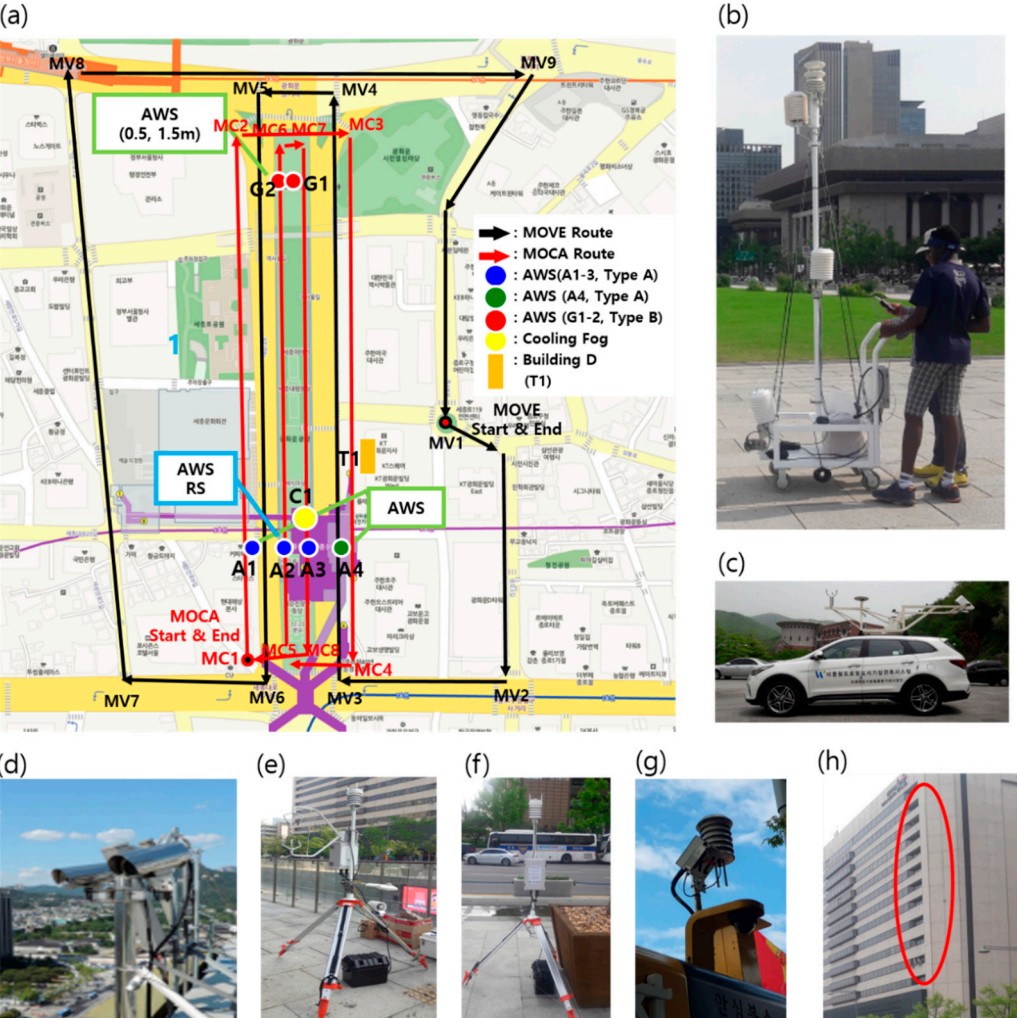

**Figure 3.** (**a**) Location of fixed instruments and sensors, and MOCA and MOVE during the BBMEX, (**b**) MOCA, (**c**) MOVE, (**d**) thermal infrared imagery, (**e**) radiation and turbulence measurement system, (**f**) integrated meteorological measurement system, (**g**) integrated meteorological sensor installed over public telephone box, and (**h**) integrated meteorological sensors deployed at 7 levels outside building D.

A Mobile Meteorological Observation Cart (MOCA) was assembled to monitor the surface and 3 levels of air temperatures on the surfaces of sidewalks and GHM squares (Figure 3b). Air temperature sensors were deployed at heights of 0.5 m, 1.5 m, and 2.5 m. A surface temperature sensor (model: SI-111-SS, manufacturer: Apogee) was deployed at a height of 0.5 m to detect radiative energy 8 to 14 µm from the field of view with 22° half angles. The data were sampled every 1 s. The operation

route circulates two sideways and GHM squares for approximately 40 min (Figure 3a). The MOCA was operated 23 times during the campaign (Table 3).

**Table 3.** Observation time for mobile vehicle and cart during the BBMEX campaign.

| No. | Mobile Vehicle (MOVE) | | Mobile Cart (MOCA) | |
|---|---|---|---|---|
| | Day | Time (LST) | Day | Time (LST) |
| 1 | | 1451–1516 | | 1200–1240 |
| 2 | | 1548–1618 | | 1300–1340 |
| 3 | 5 Aug 2019 | 1758–1818 | | 1400–1440 |
| 4 | | 1948–2017 | 5 Aug 2019 | 1500–1540 |
| 5 | | 2048–2115 | | 1600–1640 |
| 6 | | 0648–0716 | | 1700–1740 |
| 7 | | 0754–0807 | | 1800–1840 |
| 8 | | 0848–0919 | | 2100–2140 |
| 9 | | 0947–1015 | | 0400–0440 |
| 10 | | 1048–1115 | | 0600–0640 |
| 11 | 6 Aug 2019 | 1148–1220 | | 0700–0740 |
| 12 | | 1447–1515 | | 0800–0840 |
| 13 | | 1547–1622 | | 0900–0940 |
| 14 | | 1647–1705 | | 1000–1040 |
| 15 | | 1747–1821 | 6 Aug 2019 | 1100–1140 |
| 16 | | 1947–2017 | | 1200–1240 |
| 17 | | 2047–2114 | | 1300–1340 |
| 18 | | | | 1400–1440 |
| 19 | | | | 1500–1540 |
| 20 | | | | 1600–1640 |
| 21 | | | | 1700–1740 |
| 22 | | | | 1800–1840 |
| 23 | | | | 2100–2140 |

A Mobile Observation Vehicle (MOVE) was installed with meteorological sensors, a road surface sensor (model: DSP101 and DSC111, manufacturer: Vaisala) and a Global Navigation Satellite System (GNSS) antenna (model: NetR9, manufacturer: Trimble) (Figure 3c; Table 2) [27]. The road surface sensor can measure road surface temperature, status (dry, wet, moist, ice, etc.), salinity, water depth, and conductivity every 1 s. The operation route includes Sejong-daero Streets and nearby roads, as depicted in Figure 3a. The MOVE operated 17 times during the campaign period (Table 3). The data obtained by MOVE are detailed in [30].

*2.3. Sky View*

The sky view of a given location has a very important role in computing incoming solar radiation in a BB and shade by buildings. Using photos taken by a fish-eye lens camera, the elevation angles of obstacles were calculated at every 1 degree of azimuth angle. A sky view factor (SVF) can be calculated using the elevation $\theta$ and azimuth $\phi$ angles of obstacles,

$$SVF = \frac{1}{2\pi} \int_0^{2\pi} \cos^2 \theta \, d\phi. \tag{5}$$

The unity SVF implies that the point is completely open to the sky without any obstacles, whereas the zero SVF means that the point is completely shielded by obstacles [31].

*2.4. Experimental Period*

The experimental campaign period was 1200 LST on 5 August 2019 to 2100 LST on 6 August 2019. The yearly highest temperature of 40.2 °C was recorded in Anseong City (south of Seoul City),

Korea, on 5 August, while the yearly highest temperature at the Seoul station of 36.8 °C was recorded between 1450 LST and 1456 LST and between 1550 LST and 1558 LST on 6 August 2019. The daily minimum temperatures were 25.6 °C and 27.9 °C on 5 August 2019 and 6 August 2019, respectively; the values exceed the criteria (25 °C) for a tropical night in Korea. Heat wave warnings (daily maximum temperature >35 °C) were issued on both days.

*2.5. Synoptic Meteorological Features*

During the campaign period, the Korean Peninsula was hot and humid mainly due to the North Pacific High. The Seoul Metropolitan Area (SMA; Seoul City, Gyeonggi Province, and Incheon City) was affected by a northwesterly wind due to the Siberian High, while the eastern regions of the SMA were affected by easterly winds due to the North Pacific High on 5 August 2019. Thus, the maximum temperature occurred on the eastern side of the SMA by the temperature increase due to the föehn effect (Figure 4b). Nonetheless, the horizontal distribution of air temperature showed a typical night-time urban heat island pattern, which shows higher temperatures in urban areas and lower temperatures in rural areas, at 0600 LST on both 5 August 2019 and 6 August 2019 (Figure 4a,c). As the North Pacific High expanded to the west and the 8th typhoon 'Francisco' moved from Japan to southern Korea, the SMA was affected by the easterly wind the next day [30].

Figure 4e shows the time series of the air temperature at the Seoul, Incheon, and Yangpyeong ASOS stations. Three stations are located at nearly the same latitude but the Incheon station is located near the shoreline, and the Yangpyeong station is located farther inland and is classified as a suburban area. The Incheon station did not show high temperatures due to the direct effect of the Yellow Sea [32]. The Yangpyeong station showed higher temperatures than the Seoul station on 5 August but the Seoul station showed higher temperatures than the Yangpyeong station since 2100 LST on 5 August. The temperature difference between the Seoul and Yangpyeong stations is considered an index of UHI intensity in the SMA e.g., [33].

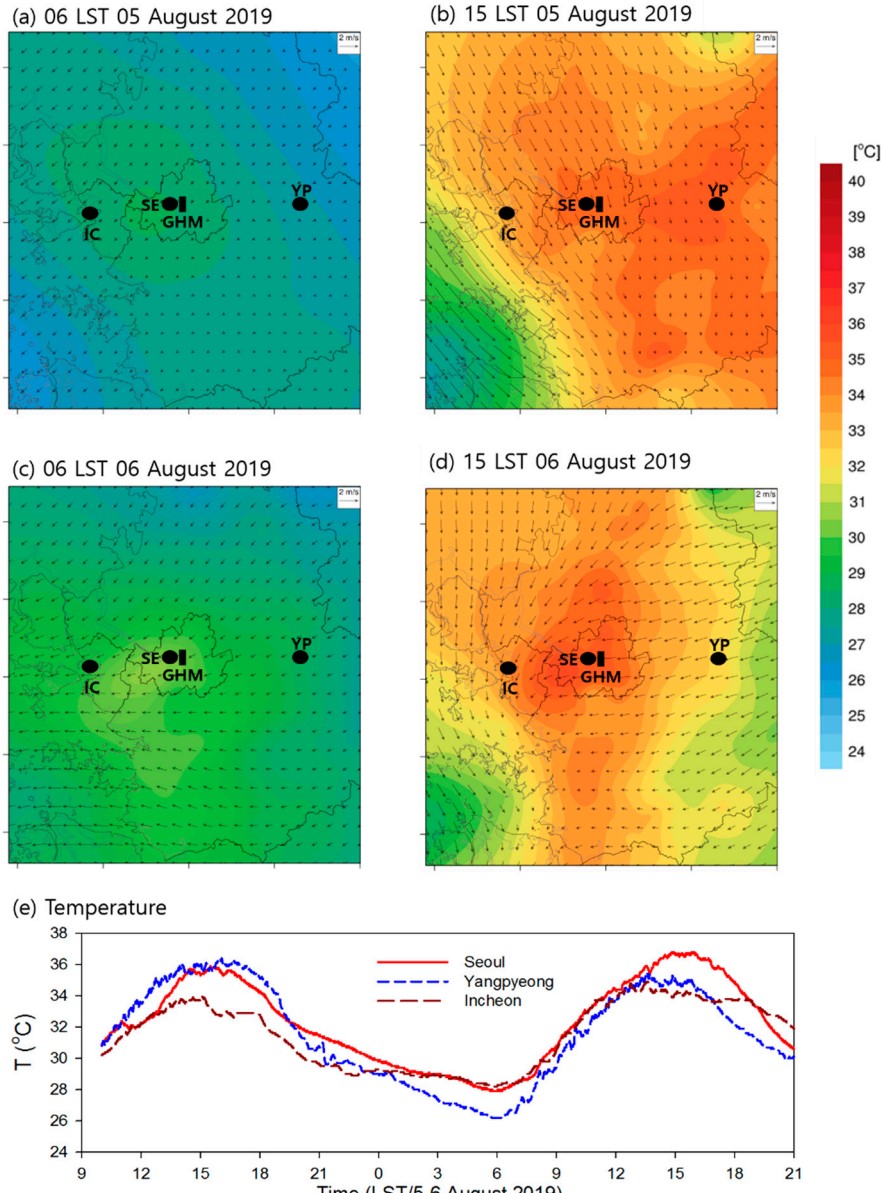

**Figure 4.** Horizontal distribution of air temperature and wind in Seoul Metropolitan Area at (**a**) 0600 LST, (**b**) 1500 LST on 05, (**c**) 0600 LST, and (**d**) 1500 LST on 06 August 2019, and (**e**) time series of air temperature observed at the Seoul (SE, red), Yangpyeong (YP, blue), and Incheon (IC, brown) automatic synoptic observation system (ASOS) stations. The location of the stations and the Gwanghwamun (GHM) campaign area are indicated by black filled circles and a rectangle, respectively.

### 2.6. Horizontal Distribution Near the Campaign Domain

Figure 5 shows the horizontal distribution of temperature obtained by the AWS network operated by kt company on 0500 LST and 1500 LST on 6 August 2019. The AWS sensors of the kt network (ktWSs) were installed at a height of 2 m (a.g.l.) above the public telephone box on sidewalks in street canyons (Figure 3g). The air temperature was higher than that at the Seoul station by 2 °C at 0500 LST but did not show considerable spatial differences among stations (Figure 5a). The GHM domain showed a slightly lower temperature than that at the southern and western parts of the domain at 1500 LST (Figure 5b). A spatial difference among the stations of 4.2 °C was observed at 1500 LST. Generally, the areas with higher temperature in daytime corresponded with those with denser BBs.

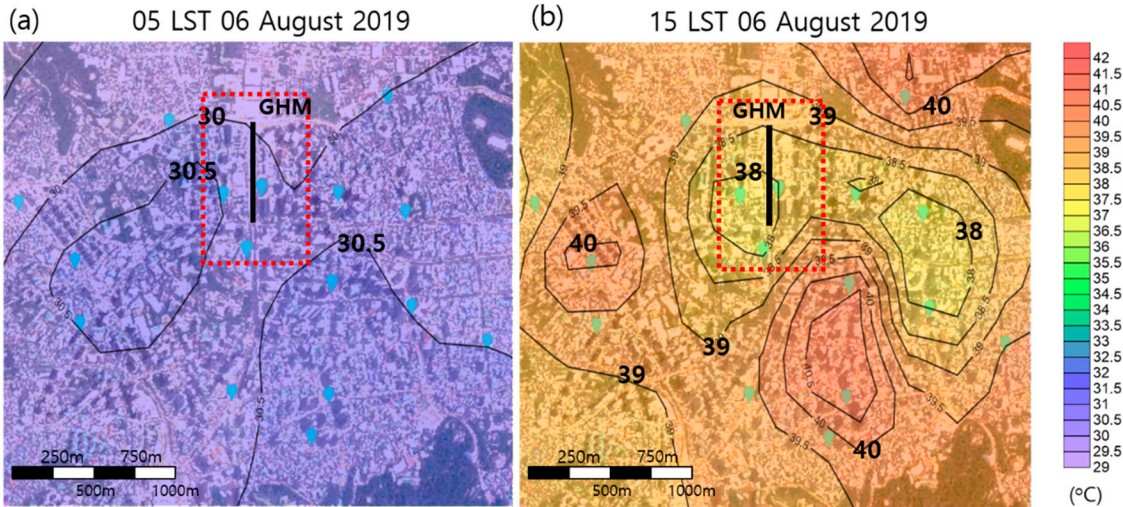

**Figure 5.** Horizontal distribution of air temperature obtained by kt automatic weather system (AWS) sensor network (indicated by light-blue or light-green symbols) at (**a**) 0500 LST and (**b**) 1500 LST on 6 August 2019. Open red dotted rectangle and closed black rectangle indicate the BBMEX campaign domain and the Gwanghwamun Square, respectively.

## 3. Results

### 3.1. Sky View and Sky View Factor

Figure 6a,b show an example of a photo taken by a fish-eye lens camera and elevation angles of obstacles with respect to the azimuth angle. The sky views at 18 locations were selected in the campaign domain (Figure 6c). The northern parts (point number 1–3) of the domain are relatively open to the sky, while the southern parts (point number 16–18) are relatively closed to the sky (Figure 6c). The middle (point number 2, 5, 8, 11, 14, 17) of the street canyon are more open than both sides of the canyon. Some points are shaded by buildings, and some points (point number 7, 10, 12, and 13) are shaded by buildings and trees. As a result, the average SVF in the middle of the street canyon was 0.69 and much higher than that for the sidewalks. The mean SVF for the western sidewalk was lower than that for the eastern sidewalk. The mean SVF in the most northern row (1–3) was the highest (0.82), while that in the fifth row (13–15) was the lowest (0.37).

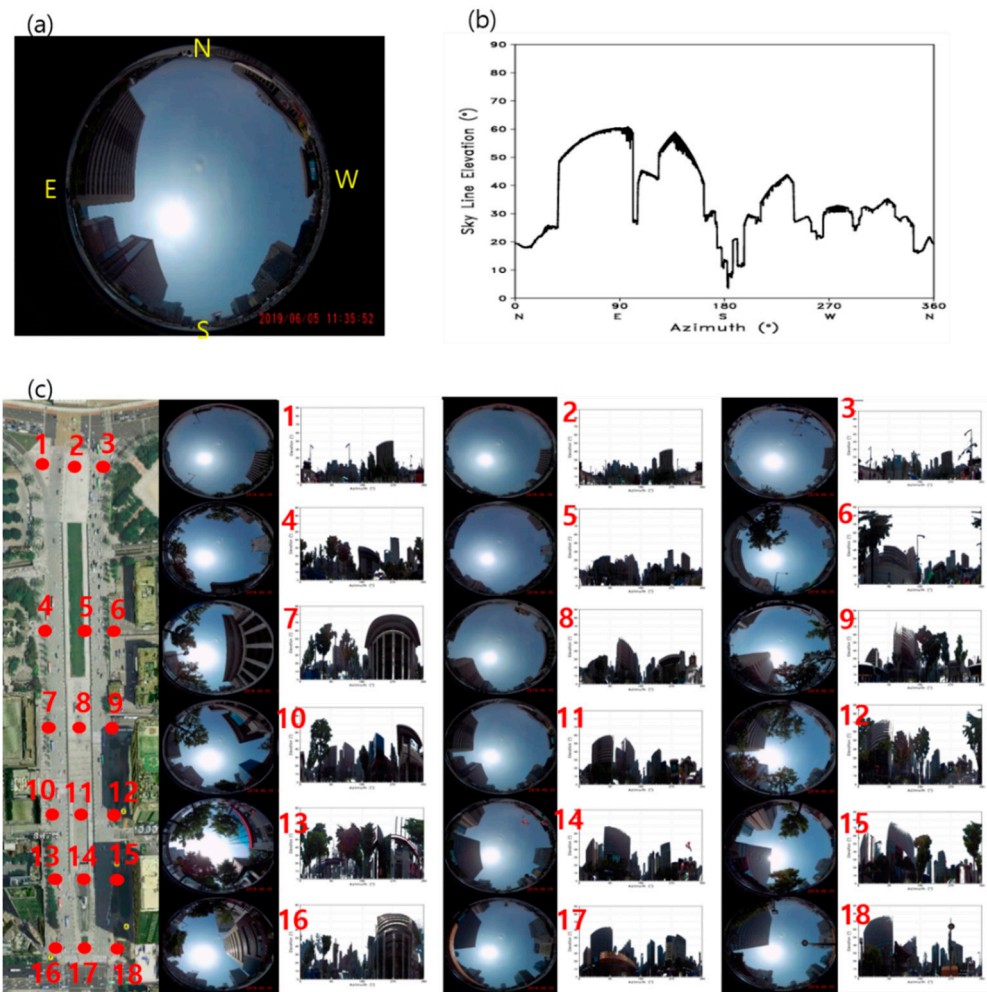

**Figure 6.** (**a**) A photo taken of the zenith by a fish-eye lens camera, (**b**) sky line elevation with respect to the azimuth angle, and (**c**) sky views and sky line elevations at 18 selected points in the campaign domain.

### 3.2. Temporal and Spatial Distribution of Surface Temperature Obtained by Thermal Infrared Imager (TIR)

Figure 7 exhibits the horizontal and vertical distributions of the surface temperature at the selected time observed by TIR on 6 August 2019. At 0750 LST, 2 h after sunrise, most surfaces showed similar surface temperatures in the range of 31−34 °C (Figure 7b). On the eastern facades of buildings (Buildings A, B, and C), the surface temperatures increased from 0830 LST, reached a maximum between 0910 LST and 1010 LST, and then decreased (Figure 7c,e). The low surface temperature area on eastern facade of Building A was due to the shade of Building G at 0750 LST and 0830 LST (Figures 1 and 6b,c). Because the solar elevation angle was large between 1100 LST and 1340 LST, the surface temperatures on horizontal surfaces, such as roads and the GHM square, were also large and those on the eastern facades of buildings decreased (Figure 7f,j).

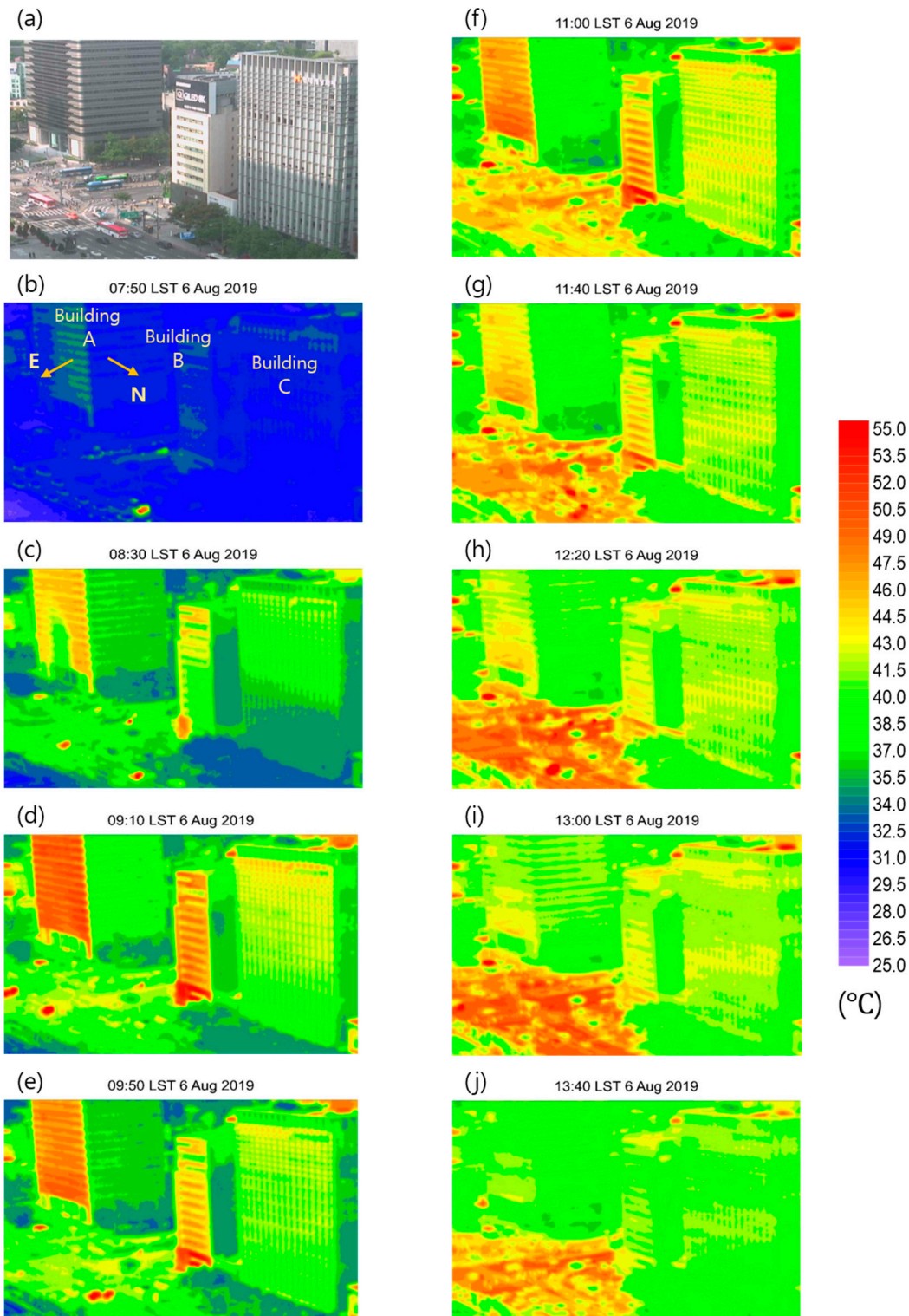

**Figure 7.** (**a**) Real image corresponding to the view angle of thermal infrared imager at 0830 LST, and horizontal distribution of surface temperature observed by TIR at (**b**) 0750 LST, (**c**) 0830 LST, (**d**) 0910 LST, (**e**) 0950 LST, (**f**) 1100 LST, (**g**) 1140 LST, (**h**) 1220 LST, (**i**) 1300 LST, and (**j**) 1340 LST on 6 August 2019.

The temporal variation in surface temperature directly depends on the solar zenith or elevation angles at a given surface, which is a function of slopes of facades and shade/sunlit by surrounding

obstacles. The facades at points 1, 3, and 4 are directed to the east, while the facade at point 2 is directed to the north (Figure 8a). Points 5 and 6 represent a tree and a road, respectively, on horizontal surfaces. The surface temperatures at points 1, 3, and 4 exhibited a similar diurnal pattern with a maximum near 0930 LST, whereas those at other points exhibited a similar diurnal pattern with a maximum near 1430 LST. These findings imply that the street canyon can be heated by the eastern facades of buildings before noon, the horizontal surfaces near noon, and the western facades after noon, respectively. These results imply that the air temperatures in the street canyon may start to increase at an earlier time than those over the horizontal flat surfaces and higher temperatures of the former persist to a later time than the latter. Due to an increase in the number of heated surfaces, the air temperatures in BBs may be higher than those over horizontal and flat surfaces.

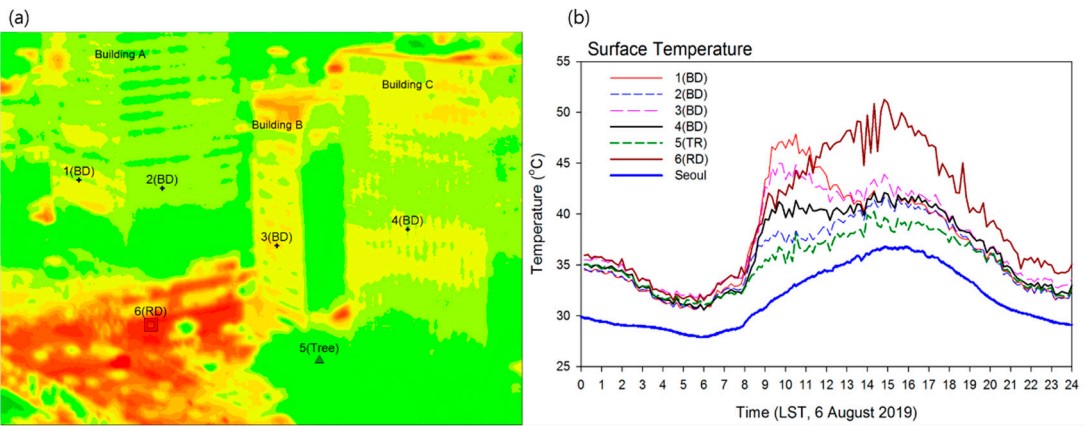

**Figure 8.** (**a**) Locations of 6 selected points, and (**b**) time series of surface temperature at the 6 selected points and the Seoul station on 6 August 2019.

### 3.3. Air and Surface Temperatures in Building Blocks Obtained by AWSs

Figure 9 shows the time series of 4-components radiative flux and turbulent parameters observed by the RS system at point A2 of GHM Square (Figure 2). The abrupt increase in the downward solar radiation at 1015 LST implies that the sun had risen above buildings at that time. The observed downward solar radiation before this time was derived from the diffused component mainly due to aerosols or the reflected component of solar radiation. Albedo has a range between 0.15 at approximately 1100 LST and 0.2 at approximately 1700 LST (Figure 9c). The outgoing longwave radiation was larger than the incoming longwave radiation all day; thus, the net longwave radiation was negative. Between sunset and sunrise, the net radiation becomes negative (Figure 9b).

Correspondingly, the friction temperature and sensible heat flux show nearly zero and slightly negative values from 1900 LST on the first day to 1015 LST on the second day (Figure 9d). The friction velocity shows smaller values of approximately 0.2 m s$^{-1}$ during night-time, and larger values of approximately 0.3–0.5 m s$^{-1}$ during daytime (Figure 9d) [34].

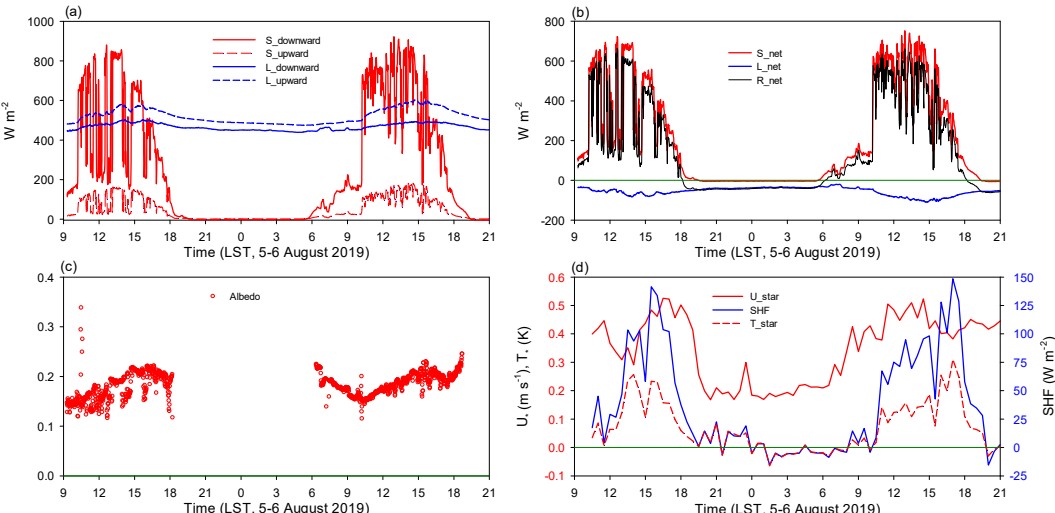

**Figure 9.** Time series of (**a**) 4-components radiative fluxes, (**b**) net shortwave, net longwave, and net global radiative fluxes, (**c**) albedo, and (**d**) friction velocity, friction temperature, and sensible heat flux observed at point A2 of Gwanghwamun Square during the BBMEX campaign period.

Figure 10 shows the time series of temperature obtained from fixed sites during the BBMEX campaign. Note that the Seoul station is located at a similar elevation as the GHM station of the rooftop of Building D. Nonetheless, the air temperature at the GHM station was higher than that at the Seoul Station by 1.1–1.9 °C at night-time, and the former was similar to the latter in daytime. The temperature inside a BB (GHM_A2) was warmer than that over the rooftop of a BB in daytime, and the former was similar to the latter at night-time (Figure 10a). Site A1 was open to the sun only during the morning time and was shaded by buildings and trees during the remaining time. Thus, GHM_A1 exhibited the lowest temperature among the AWSs in the street canyon except for the morning (Figure 10b). Maximum difference of air temperature between A1 and A2 was 0.9 °C at 1042 LST, while that between A4 and A3 was 2.6 °C at 1535 LST (Figure 3a). On the other hand, GHM_A4 exhibited the highest temperatures in the afternoon due to the heat transfer from not only the underlying surfaces but also the western facades of Buildings D and E (Figure 10b). Because the face of Building D has sensors directed to the west, the temperatures exhibited the highest values at a height of 14 m and decreased with an increase in the heights of the sensors (Figure 10d). A higher temperature at the middle height of Building D implies that the heat, might have been released from ventilators for air conditioning in daytime.

The temperature over the grass surface (G1) was lower than that over the pedestrian path (G2) throughout the day, especially at a height of 0.5 m (Figure 10c), which might be due to the large latent heat flux from grass surfaces.

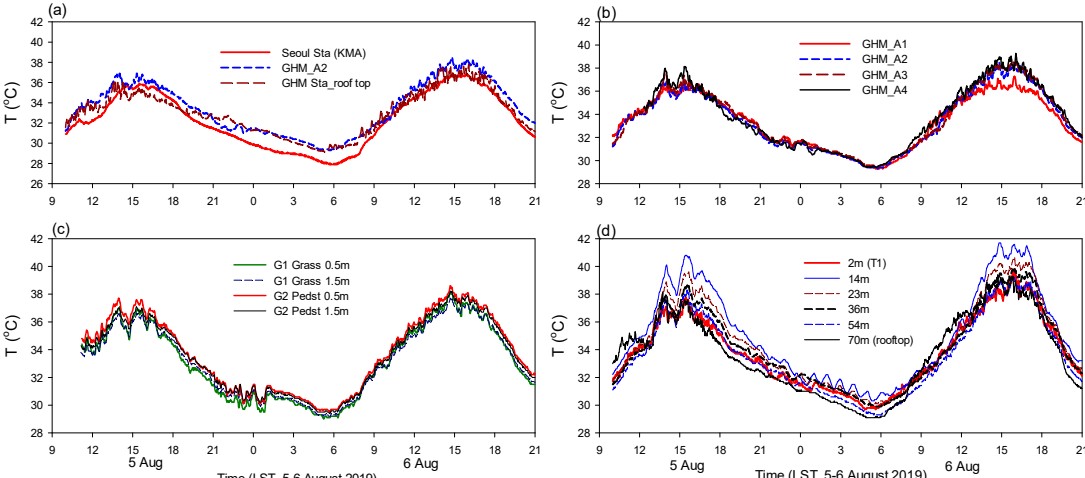

**Figure 10.** Time series of air temperatures observed at (**a**) the Seoul ASOS Station, GHM_A2, GHM station (rooftop of Building D), (**b**) GHM_A1, A2, A3, and A4, (**c**) GHM_G1 and G2 at heights of 0.5 m and 1.5 m, and (**d**) heights of T1, 14 m, 23 m, 36 m, 54 m, and 70 m outside Building D during the BBMEX campaign period.

Figure 11a,b show the time series of the surface and air temperature obtained by MOCA at 0600 LST and 1600 LST on 6 August 2019. The 0600 LST and 1600 LST were chosen as typical times with the lowest and hottest temperatures, respectively. Figure 11c,d show the photos taken at the sidewalk with trees, near the ground fountain, and on the road, respectively. The minimum spatial variation in the range of surface temperature was 4 °C at 0600 LST, while the maximum spatial variation was 20 °C at 1600 LST. Abrupt increases and decreases in surface temperature on the route from MC1 to MC2 at 1600 LST showed that the surface temperatures were directly affected by sunshine or shade in daytime (Figure 11b). For example, the surface temperature at the sidewalk with trees (Figure 11e) and near the ground fountain (Figure 11f) exhibited lower values than the surrounding surfaces by approximately 10 °C at 1600 LST (Figure 11b,d). The roads exhibited higher surface temperatures in daytime (Figure 11b,d,g). Spatial variations in the surface temperature at 0600 LST indicates that a lower or higher surface temperature in daytime could persist until the next early morning (Figure 11b).

The air temperature at 1.5 m height was higher than that at 2.5 m by 0.5 °C on the GHM Square at 1600 LST (Figure 11b,c). The minimum spatial variation in the range of air temperatures was 0.2 °C at 0600 LST, while the maximum spatial variation was 1.5 °C at 1600 LST (Figure 11a–c). Neglecting small variations, air temperatures increased on the routes from MC3 to MC4 and from MC7 to MC8 and decreased on the route from MC5 to MC6 (Figure 11c). These implied that higher air temperatures corresponded to larger surfaces with western facades of buildings in a BB, and lower temperatures corresponded to smaller surfaces with western facades, by and large, in daytime. Around 0.5 °C of decrease in air temperature at 1616 LST and 1619 LST mainly resulted from the shade by trees.

The historic waterway was designed to make water flow on hot summer days (Figure 2d). Water flowed through the historical waterway with a depth of 2 cm from 1000 LST to 1800 LST on 5 August 2019. The surface temperatures on two points with the same material were observed by a portable infrared (IR) thermometer (model 62 mini, manufacturer Fluke) at 0430 LST, 0715 LST, and 0950 LST on 6 August 2019. Two points were located at a distance of 0.3 m, one point was located in the dried waterway, and one point was located outside the dried waterway. The surface temperatures on the former were 26.6 °C, 31.6 °C, and 35.6 °C, and those on the latter were 29.6 °C, 33.2 °C, and 36.8 °C, respectively. These differences imply that the effect of a waterway on mitigating the surface temperature can continue to the next morning.

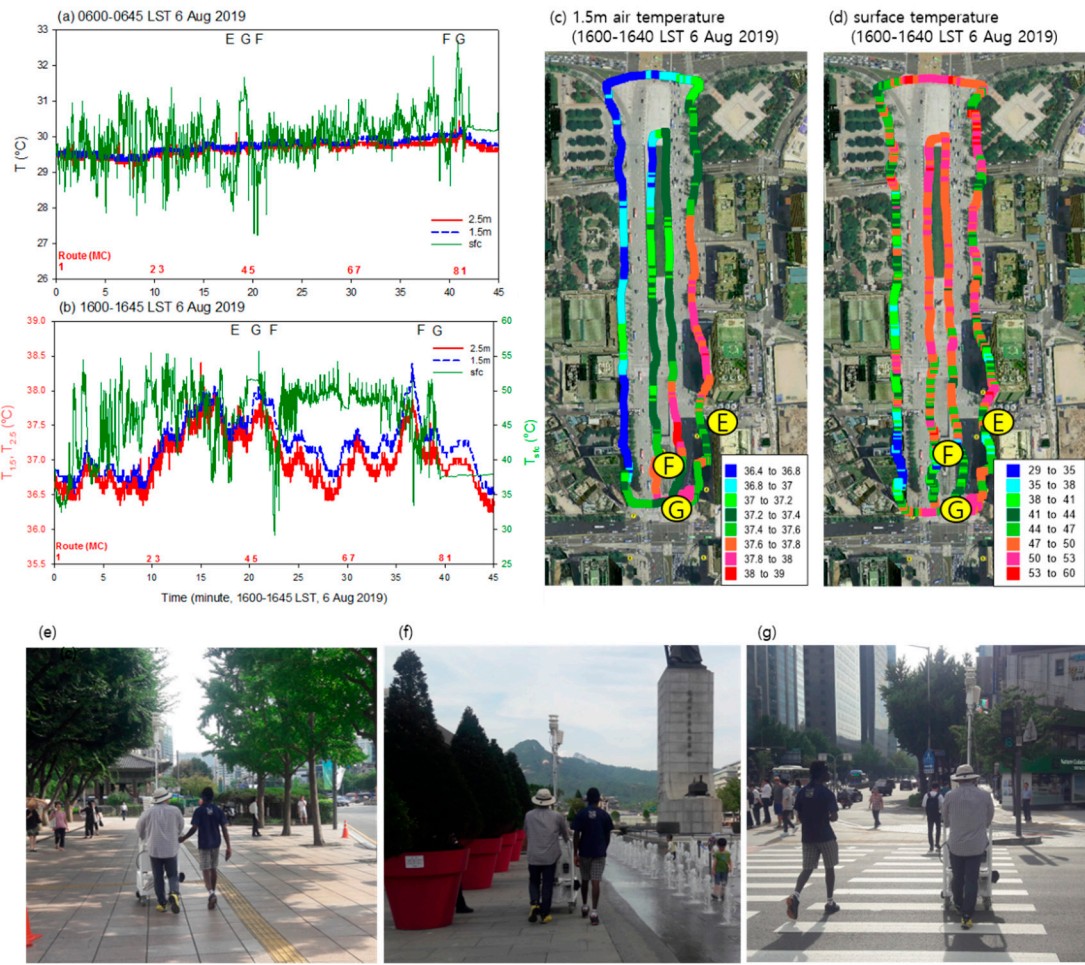

**Figure 11.** Time series of surface, 1.5 m, and 2.5 m air temperatures obtained by MOCA during (**a**) 0600-0650 LST, and (**b**) 1600–1650 LST on 6 August 2019. Horizontal mapping of (**c**) 1.5 m air temperature and (**d**) surface temperature on satellite image of BBMEX domain. Landscapes at (**e**) point E (tree), (**f**) point F (ground fountain), and (**g**) point G (road) in (a) and (b). Route number in (a) and (b) indicates the location with its heading MC in Figure 3a.

## 4. Summary and Discussion

### 4.1. Summary

The 2019 BBMEX campaign was carried out in the GHM area during the heat wave and tropical night periods (5 to 6 August) in 2019. The campaign was organized by the National Institute of Meteorological Sciences and Hankuk University of Foreign Studies, and was participated by SIT, kt, Kongju University, and Observer Inc. The campaign period was 1200 LST on 5 August 2019 to 2100 LST on 6 August 2019. The period includes the highest temperature records in Korea on 5, and Seoul on 6 August 2019, respectively. The climate on these days was very hot and humid due to the North Pacific High. Before the campaign, the SVFs were calculated using the sky views at 18 points in the campaign domain.

Many meteorological instruments were integrated and installed in the domain. A TIR observed the surface temperature with 320 × 240 pixels, including eastern facades of three buildings and main roads, every 10 min. Considerable increases in surface temperature on the eastern facades of buildings before noon and on horizontal surfaces near noon were observed.

The air temperatures on the second day inside the BB were higher than those at the Seoul ASOS station by 0.1−2.2 °C in daytime, and 1.1−1.9 °C in night-time. The AWS station at eastern (western)

sidewalk in the BB showed the highest temperature in the morning (afternoon). The air temperature near Building D exhibited a maximum at a lower height (approximately 2/7 of building height in this study) in this BB and decreased with height in daytime. These were mainly due to the heat releases from ventilators in the BB. The surface and air temperatures in the BB were relatively homogeneous at night-time, but were substantially inhomogeneous in daytime. The meter-scale surface and air temperatures in the domain were observed 23 (17) times on the sidewalks (roads) using the MOCA (MOVE). Sidewalks with trees and sites near the ground fountain exhibited lower temperatures while roads and open pavement exhibited higher temperatures. The surface temperature was found to be directly affected by sunlit or shade and solar zenith angle to the surface in daytime.

*4.2. Discussions*

Through the BBMEX campaign, many valuable datasets have been acquired to quantify the difference of surface and air temperatures between outside of and in a BB as well as among different points in the BB. These datasets will contribute to quantifying the horizontal and vertical distribution of surface and air temperature in a BB. Nonetheless, the BBMEX datasets have some limitations for making generalizations, due to their short period and limited areas. Much more datasets with longer period in the same BB or in other BBs are necessarily needed.

Data processing in a domain with a meter-scale inhomogeneity has some weaknesses. For an instance, the eddy covariance method needs flat and homogeneous surfaces with a sufficiently distant fetch [35]. But, it is impossible to find ideal sites for the analysis in a BB. Also, the surface temperature was calculated from radiative flux obtained by TIR by assuming that emissivity for all surfaces is 1. If a surface has an emissivity of 0.5, the estimated surface temperature can differ from the real one by above 40 °C. The exact emissivity for each pixel could be determined by comparing the contact-type and the infrared-type surface temperatures.

As mentioned earlier, the high-resolution temperature field in a realistic BB is dependent on many factors: the structure of BB (e.g., direction of main building array, aspect ratio of street canyon), the thermal properties of surface material (e.g., thermal conductivity, heat capacity, specific heat), and physical properties of the surface (e.g., albedo, emissivity). Moreover, incident solar energy, energy partition (e.g., heat storage, sensible and latent heat flux), energy transfer (into the surface materials or between the surface to atmosphere) in vertically as well as in horizontally should be considered.

Overall, the available solar energy at a given point is proportional to SVF (strictly, sum of cosine of solar zenith angle for the period from sunrise to sunset). However, BBs have larger surfaces than the horizontal flat surfaces. As a result, a BB could receive more energy than the horizontally flat surfaces due to its smaller SVF, whose surplus energy might depend on direction of a building array and aspect ratio of the street canyon, and so on. The higher temperature might be due to the heating on the eastern facades of the buildings before noon, on horizontal surfaces near noon, and on western facades of buildings after noon. Eventually, the greater solar radiation in the BB could result from the larger surface areas.

In order to apply these BBMEX datasets to BB-scale meteorological models such as a large eddy simulation (LES) or direct numerical simulation (DNS), more sophisticated energy transfer process between wall of a building and air inside the BB should be parameterized on the basis of observation and verified e.g., [36]. In order to improve the dynamical processes in a BB, a high-resolution turbulence observation campaign needs to be designed and carried out in near future. It is because that energy transfer between a surface and air depends on not only temperature difference, but also turbulent quantities such as friction velocity. Also, the ratio of direct solar radiation to global solar radiation could be a necessary factor to parameterize the incoming solar radiation.

These days, many local governments in Korea have tried to expand facilities mitigating heat stress in urban areas in various ways: planted trees, cooling fogs/mists, clean roads, ground fountains, and waterways [37]. In order to generalize the effects of these facilities on temperature decrease, more datasets should be acquired and analyzed in terms of diverse environmental factors. All results could

be applicable to short-term or long-term urban planning and renewal. These could give a guideline to determine the best direction of a building array, the best aspect ratio of a street canyon to minimize heat stresses in BBs, as well as the type, number, and location of mitigating facilities. The BBMEX dataset will be a milestone to make cities more resilient and more sustainable.

**Author Contributions:** Conceptualization, M.-S.P. and S.-H.L.; methodology, M.-S.P., J.-Y.B., B.-J.K., W.C., K.-M.M., T.-I.C.; BBMEX design and site survey, M.-S.P., J.-H.C., J.-S.M., S.-H.K., C.-R.C.; BBMEX instrumentation, M.-S.P., W.C., K.-M.M., T.-I.C., J.-H.C., J.-S.M., S.-H.K., C.-R.C.; BBMEX weather forecast, S.-H.K., C.-R.C.; BBMEX civil permission, W.C., K.-M.M., M.K.; BBMEX sky-view investigation, M.K., J.-B.J.; BBMEX staff, J.-S.M., M.K.; Gwanghwamun Station, J.-Y.B.; kt sites, K.-M.M.; MOVE operation, B.-J.K.; MOCA assemble and operation, M.-S.P., J.-H.C., M.K.; manuscript preparation and writing, M.-S.P.; funding acquisition, M.-S.P., S.-H.L. All authors have read and agreed to the published version of the manuscript.

**Funding:** This research was funded by the Korea Meteorological Administration Research and Development Program under Grant KMI 2018-05612.

**Acknowledgments:** The Gwanghwamun UMS-Seoul data were supported by the Korea Meteorological Administration's National Institute of Meteorological Sciences Development of Advanced Research on Biometeorology and Industrial Meteorology (1365003004). Authors thank all attendants and assistants (K.-H. Kwak and his colleagues, S.-H. Lee and his colleagues, J.-Y. Byon and his colleagues, B.-J. Kim and his colleagues, C. Lee and H.-J. Yang from the Research Center for Atmospheric Environment, Hankuk University of Foreign Studies, who pushed MOCA, and drove the vehicle MOVE) during the hot summer campaign period. The authors also thank kt company for providing several facilities, including the parking lot for MOVE and meeting lounge.

**Conflicts of Interest:** The authors declare no conflict of interest.

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
