# Peer review of "A Building-Block Urban Meteorological Observation Experiment (BBMEX) Campaign in Central Commercial Area in Seoul"

_atmosphere, doi:10.3390/atmos11030299_

Round 1
Reviewer 1 Report
This study is about an intensive meteorological observation campaign in Seoul, Korea during two days with heat wave conditions. The measurement results are overall well presented. However, the real asset of this study is not clarified. You have done an intensive microscale observation campaign during heat wave conditions in a dense building block with several heat mitigation implementations next to each other: grass, water fountains, fog nozzles and tree shade. And then the central area and one end of the square do not seem to have any mitigation options, i.e. it is just bare concrete which seems to be a perfect situation for comparison with the areas with mitigation options. This is to me the real value of this study, i.e. getting insights into the empirical evidence of heat mitigation options and how they perform in comparison. Currently the manuscript only scratches the surface of this potential. I hope you make improvements so that the reader can extract this valuable information (cooling potential of several heat mitigation options).
For this reason and further remarks given below, the manuscript is suitable to be published in Atmosphere after major revisions.
Remarks:
Introduction:
The introduction does not clarify the motivation of this study. The first 5 lines of the discussion section do highlight the motivation and also the originality (micro-β to γ scale observations). This needs to be moved to the introduction. And I also think there a lot of microscale observation studies that could be mentioned as well for comparison, although the scale concerning the amount of instruments involved here is indeed remarkable. You should further mention which kind of research questions you hope to answer with this campaign.
Discussions:
The discussion section is currently not a discussion but just a summary of the results. This needs to be improved e.g. by setting your results in the context of past observation campaigns focusing on heat waves in dense urban areas. What are the new aspects? What is in line with or contradicting past observations? Can this dataset be used for certain models for validation purposes? These are questions that should be answered in the Discussion section and a summary of the major results can be given in concluding paragraph (or Conclusions section).
You also have to discuss the uncertainty of the surface temperatures due to unknown emissivities of the materials. Just suppose a material emits e.g. 500 W/m2 and an emissivity of 1 is assumed but the real emissivity is rather low with 0.85 (e.g. weathered steel), then the error in the surface temperature estimation is about 12-13K.
Detailed remarks:
l. 40: change “4.2 billion over 3%” to “4.2 billion and urbanized regions cover over 3%” or similar
l. 42: change “complex comparison” to “complex pattern in comparison” or similar
l. 71: which scale is this study covering? You mention it in the beginning of your discussion section but it should be mentioned here first.
l. 78: dot missing at end of sentence
l. 81: what is the overarching purpose for conducting this campaign?
l. 100: “waterways of history” - this is confusing to me since I would think this means “historic waterways” but looking at Fig 2d they seem to be relatively new…
l. 101: is “Sejong-daero” the name of the street or does this have a different meaning?
l. 102: change “Each sidewalk” to “A sidewalk”
l. 113: change “over” to “on”
l. 130: change “deploys” to “was used with” or similar
l. 138-139: I assume you mean “friction temperature” instead of “fiction temperature”
l. 152: point to Figure 3 here. I was trying to find points G1 and G2 in Figure 2.
Table 2: A complete list of the instrumentation is necessary including model and accuracy of the instrument
l. 192: change “by the easterly” to “by easterly”
l. 222: according to Fig 5 it is rather 2.5 °C
Figure 5: Where are the stations of the network? What kind of interpolation was used?
l. 231: change “A sky view” to “The sky view”
l. 235: the calculation of the sky view factor has to be presented in the Material and Methods section, not in the results section
ll. 271-272: I assume it should be possible to test this with a weather stations in the surroundings of Seoul
Figure 7: if available a photograph with the same or similar field of view would be helpful so that the reader gets a sense of the materials involved
Figure 9 a: change “S_downard” to “S_downward”. Also the terms given in the legend should be explained in the Figure caption.
Figure 9: where was the location of this measurement?
l. 300: How is the Seoul station situated? Urban or vegetated surroundings? What is the surface material below the station?
l. 305-306: this is an interesting insight. What was the max temperature difference to unshaded locations?
l. 310: could it be influenced by waste heat from air conditioning?
l. 321-323: you should state here that you only show results of 2 out of 23 MOCA walks and clarify why you chose 6 and 16 LST
l. 333: change “1.5 m high” to “1.5 m height”
l. 337-338: I do not think that SVF is a good explanation here since then you would need to see the same trend from MC1-2 which is not the case. It might rather be influenced by the hot western facades of buildings D and E.
l. 349: last sentence in that line should be mentioned in the Material and Methods section, not here
l. 371: “a building block” - I think you should clarify that you mean your study area which you could also abbreviate with BB e.g. (throughout the manuscript)
l. 372: “eastern faces” should be “eastern facades” I think
l. 374: English needs to be improved here
l. 377: still wondering if this is related to waste heat
Author Response
Response to reviewer #1 comments
Authors gratefully thank to reviewer #1 for his/her thorough reviews and valuable comments which would contribute to improve the manuscript. Authors have revised the manuscript substantially to respond the reviewers’ comments. The reviewers’ all comments are responded and taken into account in the revised manuscript. In addition, many grammatical errors are corrected and incoherent sentences are rewritten throughout the manuscript. Reviewers’ comments are marked in black and the authors’ corresponding responses are marked in blue in the response. Major changes are marked in red in the revised manuscript. I hope that this manuscript will be accepted for the publication in Atmosphere.
Introduction:
The introduction does not clarify the motivation of this study.
- The motivation of this study is more clarified by adding research questions to be answered on L76-80), and several micro-scale observation experiments (L68-74) in the revised manuscript.
The first 5 lines of the discussion section do highlight the motivation and also the originality (micro-β to γ scale observations). This needs to be moved to the introduction.
- The first 5 lines of the discussion section is move to the introduction section (L56-57, L66-68).
And I also think there a lot of microscale observation studies that could be mentioned as well for comparison, although the scale concerning the amount of instruments involved here is indeed remarkable.
- Several micro-scale observation in an urban areas are added on L68-74 in Section 1. As examples, the LUCE (Lausanne Urban Canopy Experiment) had installed 92 stations over 300m×400 area from October 2006 to April 2007 to determine the sensible heat flux according to the surface geometry and type in Switzerland (Nadeau et al., 2009). And the USscan network had installed 200 sensors over 250m×430m area from July to August 2007 to investigate the fine-gridded temperature pattern in Tokyo downtown (Thepvilojanapon et al., 2010). Recently, the 52 stations in a block with 500m×700m had been installed during two sunny days in summer to verify the precinct ventilation performance (He et al., 2019).
You should further mention which kind of research questions you hope to answer with this campaign.
- The research questions to be answered are added on L76-80 in the revised manuscript. For the purpose of advancing to a meter-scale meteorology, more observation datasets are still needed. These datasets could be used to quantify the air temperature difference between at several points in a building block and at the nearest synoptic weather station from the BB, to quantify the effects of materials and structures of BB on surface and air temperature in the BB, and to quantify the effects of facilities mitigating heat stresses on the temperatures in the BB.
Discussions:
The discussion section is currently not a discussion but just a summary of the results. This needs to be improved e.g. by setting your results in the context of past observation campaigns focusing on heat waves in dense urban areas. What are the new aspects? What is in line with or contradicting past observations? Can this dataset be used for certain models for validation purposes? These are questions that should be answered in the Discussion section and a summary of the major results can be given in concluding paragraph (or Conclusions section).
- Discussion section is improved by adding the limitation of current state experiment (L428-433), applicability to LES model improvement (L440-442), necessity of more experiments in various BBs (L423-427), and applicability to urban planning and renewal for sustainable and resilient cities (L443-447).
- The detailed analysis on the temporal and spatial distribution of surface and air temperatures are now comparing with those computed by physical models.
You also have to discuss the uncertainty of the surface temperatures due to unknown emissivities of the materials. Just suppose a material emits e.g. 500 W/m2 and an emissivity of 1 is assumed but the real emissivity is rather low with 0.85 (e.g. weathered steel), then the error in the surface temperature estimation is about 12-13K.
- The weaknesses in data process are discussed on L428-433 in the revised manuscript. The eddy covariance method needs flat and homogeneous surfaces with a sufficiently distant fetch (Aubinet et al., 2012). But it is impossible to find ideal sites for the analysis in a BB. Also, the surface temperature was calculated from radiative flux obtained by TIR by assuming that emissivity for all surfaces is 1. If a surface material has an emissivity of 0.5, the estimated surface temperature can differ from the real one by above 40°C from the true one. The exact emissivity for each pixel should be taken into account.
Detailed remarks:
- 40: change “4.2 billion over 3%” to “4.2 billion and urbanized regions cover over 3%” or similar
- “4.2 billion over 3%’ is changed to “4.2 billion over 3% urbanized region” on L40 in the revised manuscript.
- 42: change “complex comparison” to “complex pattern in comparison” or similar
- “complex comparison’ is changed to “complex pattern in comparison” on L42 in the revised manuscript.
- 71: which scale is this study covering? You mention it in the beginning of your discussion section but it should be mentioned here first.
- The micro-γ scale meteorology is focused on L76-82 in the introduction.
- 78: dot missing at end of sentence
- Period is added on L87 in the revised manuscript.
- 81: what is the overarching purpose for conducting this campaign?
- The purpose of this study is rewritten on L88-92 in the revised manuscript. This study aims to introduce experimental outlines of the 2019 BBMEX campaign. The obtained surface and air temperatures dataset is to be overviewed with a view to the structure of BB, facilities mitigating heat stresses, and environmental factors. The usefulness of the collected dataset for improving micro-gamma scale meteorology, and applicability to urban planning for attaining sustainable and resilient cities are demonstrated.
- 100: “waterways of history” - this is confusing to me since I would think this means “historic waterways” but looking at Fig 2d they seem to be relatively new…
- The term is unified as ‘historic waterway’ on L114-115 in the revised manuscript.
- 101: is “Sejong-daero” the name of the street or does this have a different meaning?
- The term ‘Sejong-daero’ is a name of a street. It is modified as ‘Sejong-dearo Street’ to more clarify it throughout the manuscript (L116, L194).
- 102: change “Each sidewalk” to “A sidewalk”
- The term ‘Each sidewalk’ is changed to ‘a sidewalk’ on L117 in the revised manuscript.
- 113: change “over” to “on”
- The term ‘over’ is changed to ‘on’ on L136 in the revised manuscript.
- 130: change “deploys” to “was used with” or similar
- The term ‘deploys’ is changed to ‘was used with’ on L148 in the revised manuscript.
- 138-139: I assume you mean “friction temperature” instead of “fiction temperature”
- The term ‘fiction temperature’ is changed to ‘friction temperature’ on L158 in the revised manuscript.
- 152: point to Figure 3 here. I was trying to find points G1 and G2 in Figure 2.
- The reference ‘Figure 3a’ is added on L176 in the revised manuscript.
Table 2: A complete list of the instrumentation is necessary including model and accuracy of the instrument
- All sensors are added in Table 2. Model name, manufacturer, and accuracy of temperature sensors are also added in Table 2.
- 192: change “by the easterly” to “by easterly”
- The term ‘by the easterly’ is changed to ‘by easterly’ on L227 in the revised manuscript.
- 222: according to Fig 5 it is rather 2.5 °C
- Maximum temperature was 41.5°C, and minimum one was 37.3°C in Figure 5b. So temperature difference was above 4.2°C (L256 in the revised manuscript).
Figure 5: Where are the stations of the network? What kind of interpolation was used?
- Stations are indicated by light-blue and light-green symbols in Figure 5. The simple Kriging method is applied to interpolate.
- 231: change “A sky view” to “The sky view”
- The term ‘A sky view’ is changed to ‘The sky view’ on L198 in the revised manuscript..
- 235: the calculation of the sky view factor has to be presented in the Material and Methods section, not in the results section
- The paragraph is moved to Section 2.3 (L197-204) in the revised manuscript.
- 271-272: I assume it should be possible to test this with a weather stations in the surroundings of Seoul
- The effect of structure of a BB (direction of main buildings and aspect ratio of street canyon) on temperature in the BB is now being investigated on the basis of physical model.
Figure 7: if available a photograph with the same or similar field of view would be helpful so that the reader gets a sense of the materials involved.
- Figure 7a is change to include a real image with the same view as thermal infrared imager. Instead, the TIR image at 1030 LST is removed in Figure 7.
Figure 9 a: change “S_downard” to “S_downward”. Also the terms given in the legend should be explained in the Figure caption.
- The label ‘S_dwonard’ is corrected to ‘S_downward’ in Figure 9.
Figure 9: where was the location of this measurement?
- The observation locate is added in Caption of Figure 9 such as ‘observed at point A2 of the Gwanghwamun Square’.
- 300: How is the Seoul station situated? Urban or vegetated surroundings? What is the surface material below the station?
- The followings are added on L121-122 in the revised manuscript: The Seoul Station is surrounded by residential area except for green belt to the east. The meteorological tower is installed over flat surface covered with short grasses.
- 305-306: this is an interesting insight. What was the max temperature difference to unshaded locations?
- The following sentence is added on L339-341 in the revised manuscript. The maximum difference of air temperature was 0.9°C between A1 and A2 at 1042 LST, while 2.6 °C between A4 and A3 at 1535 LST on 6 August 2019 (Figure 3a). The sentence is added on L336-338.
- 310: could it be influenced by waste heat from air conditioning?
- The sentence is rewritten as followings on L343-344: The heat, might have been released from ventilators for air conditioning, could have been trapped inside the BB in daytime.
- 321-323: you should state here that you only show results of 2 out of 23 MOCA walks and clarify why you chose 6 and 16 LST
- The reason is explained on L358-359 in the revised manuscript. The 0600 LST and 1600 LST were chosen as typical times with the lowest and hottest temperatures, respectively.
- 333: change “1.5 m high” to “1.5 m height”
- The term ‘high’ is changed to ‘height’ on L370 in the revised manuscript.
- 337-338: I do not think that SVF is a good explanation here since then you would need to see the same trend from MC1-2 which is not the case. It might rather be influenced by the hot western facades of buildings D and E.
- The expression is changed to the followings on L374-378: higher air temperatures corresponded to larger surfaces with western facades of buildings in a BB, and lower temperatures corresponded to smaller surfaces with western facades, by and large, in daytime. Ironically, building block could receive more energy due to its smaller sky view factor, whose surplus energy depends on direction of the building array and aspect ratio of the street canyon.
- 349: last sentence in that line should be mentioned in the Material and Methods section, not here
- The sentence is moved to L196 in Section 2.2.
- 371: “a building block” - I think you should clarify that you mean your study area which you could also abbreviate with BB e.g. (throughout the manuscript)
- Building block is abbreviated with BB throughout the manuscript.
- 372: “eastern faces” should be “eastern facades” I think
- The term ‘eastern faces’ is changed to ‘eastern facades’ throughout the manuscript.
- 374: English needs to be improved here
- The sentence is rewritten.
- 377: still wondering if this is related to waste heat.
- Some explanation is added on L346-347 and 414 in the revised manuscript. The heat, might have been released from ventilators for air conditioning, could have been trapped inside the BB in daytime.
- These were mainly due to the heat releases from ventilators in a BB and their trapping inside the BB.

Reviewer 2 Report
This paper reports urban meteorological observation experiment, which is an up to date and relevant issue. It reveals how the build environment influences the development of temperature during a heat wave.
Although the paper is already well written, I have a major concern about missing scientific discussion. Please complete the paper with the reflection on the method, and scientific discussion of the results and their meaning to finish the publication. It would be very valuable to have also an answer for heat mitigation/adaptation strategies based on your results (short term, small interventions but also long term design principles for heat resilient urban environments).
I have also some minor remarks as well:
Lines 55-59: This paragraph is unclear and the relation to the previous and following one are unclear.
Table 1: A column with Resolution/spatial precision of the experiments would complete the information.
I am not sure, do I understand it well that there is a waterway with a depth of 2 cm? (Line 100)
A table with the sensor points could complete Figure 1 and the description in section 2.1 giving a comprehensive overview of the experiment design. You could combine it somehow with the information in the table 2, but I let it up to you how and if you want to do that.
Figure 2: location of the pictures (b-d) in the aerial view (a) is not recognizable.
Lines 223-224: “denser building blocks” – the readers could like to know how the density was classified, what is the difference between the densities in the areas. (maybe an approximation at least?)
Figure 7: very informative, very good and clear graphic representation!
Lines 356-358: for what exactly is it essential? How does it help the city to become more resilient or sustainable? As for now the paper delivers an impression that just meteorological measurements make cities more resilient and sustainable. This is not enough, this is just data collection.
Are there any differences of the colors of the front walls of the buildings? Any relevant differences in the surface?
Author Response
Response to reviewer #2 comments
Authors gratefully thank to reviewer #2 for his/her thorough reviews and valuable comments which would contribute to improve the manuscript. Authors have revised the manuscript substantially to respond the reviewers’ comments. The reviewers’ all comments are responded and taken into account in the revised manuscript. In addition, many grammatical errors are corrected and incoherent sentences are rewritten throughout the manuscript. Reviewers’ comments are marked in black and the authors’ corresponding responses are marked in blue in the response. Major changes are marked in red in the revised manuscript. I hope that this manuscript will be accepted for the publication in Atmosphere.
Although the paper is already well written, I have a major concern about missing scientific discussion.
Please complete the paper with the reflection on the method, and scientific discussion of the results and their meaning to finish the publication.
- Discussion section is improved by adding the limitation of current state experiment (L428-433), applicability to LES model improvement (L440-442), necessity of more experiments in various BBs (L423-427), and applicability to urban planning and renewal for sustainable and resilient cities (L443-447).
- The detailed analysis on the temporal and spatial distribution of surface and air temperatures are now comparing with those computed by physical models.
It would be very valuable to have also an answer for heat mitigation/adaptation strategies based on your results (short term, small interventions but also long term design principles for heat resilient urban environments).
- Effect of the facilities for mitigating heat stresses on temperature decreases are summarized and discussed on L443-447 in the revised manuscript. These days, many local governments in Korea have tried to expand facilities mitigating heat stresses in urban areas in various ways: planted trees, cooling fogs/mists, clean roads, ground fountains, and waterways. In order to generalize the effects of these facilities on temperature decrease, more datasets should be acquired and analyzed in terms of diverse environmental factors. All results could be applicable to short-term or long-term urban planning and renewal.
I have also some minor remarks as well:
Lines 55-59: This paragraph is unclear and the relation to the previous and following one are unclear.
- The paragraph is rewritten on L56-65 in the revised manuscript.
Table 1: A column with Resolution/spatial precision of the experiments would complete the information.
- Several micro-scale observation experiments are added in Table 1. The available resolution of experiments are added in Table 1.
I am not sure, do I understand it well that there is a waterway with a depth of 2 cm? (Line 100)
- It is right that the depth of the waterway is just 2 cm. It is not a natural, but a very shallow manual one.
A table with the sensor points could complete Figure 1 and the description in section 2.1 giving a comprehensive overview of the experiment design. You could combine it somehow with the information in the table 2, but I let it up to you how and if you want to do that.
- All instruments are included in Table 2. Model name and manufacturer as well as accuracy of temperature sensors are added in Table 2.
Figure 2: location of the pictures (b-d) in the aerial view (a) is not recognizable.
- The location of the pictures are indicated in Figure 2a.
Lines 223-224: “denser building blocks” – the readers could like to know how the density was classified, what is the difference between the densities in the areas. (maybe an approximation at least?)
- Denser building implies higher plane area density of buildings. Plane area density is defined as the ratio of plane area of building to the flat surface area.
Figure 7: very informative, very good and clear graphic representation!
- But, Figure 7a is change to include a real image with the same view as thermal infrared imager. Instead the TIR image at 1030 LST is removed in Figure 7.
Lines 356-358: for what exactly is it essential? How does it help the city to become more resilient or sustainable? As for now the paper delivers an impression that just meteorological measurements make cities more resilient and sustainable. This is not enough, this is just data collection.
- This results could apply to urban planning or renewal for attaining the sustainable and resilient cities. The followings are added on L447-449 in the revised manuscript. This study could give a guideline on the best direction of building array, the best aspect ratio of street canyon (road width and building heights) to minimize the heat stresses in BBs, and type, number, and location of mitigation facilities.
Are there any differences of the colors of the front walls of the buildings? Any relevant differences in the surface?
- The real image is inserted in Figure 7a. Detailed analyses on the effect of color and material of surface on surface temperatures are now on hands.

Reviewer 3 Report
Overall, the paper titled 'A Building-Block Urban Meteorological observation Experiment (BBMEX) campaign in central commercial area in Seoul' is potential to be an interesting paper investigating the thermal environment around a central commerical area. However, in its current form, this paper can be improved in the following aspects.
(1) Abstract, the current version of abstract is not comprehensive to inform readers with sufficient information. It should be rewritten to follow the background, research gap, research aim and objectives, the field work, findings and significance. The information in line 25-28 is not required.
(2) Introduction, the current version does not make sense to me, especially the research aim presented in the last paragraph is not to support scientific research but a practical issue. Therefore, authors are suggested restructuring the introduction to evidence a scientific question. You may start from line 55, to tell people that the existing monitoring system at the block, precinct, neighbourhood level is not supported by the high-resolution network and what is the limitation of them? And then you can tell people the significance of high-resolution monitoring network. However, the existing networks (Table 1) cannot necessarily support the local climate studies. "And you aim is to develop a local-scale high-resolution monitoring system and to test its applicability. Moreover, the final purposes should be understanding the local thermal environment."
(3) Figure 4, you may add the wind information, as both surface and air temperatures are sensitive to areodynamics properties.
(4) The results have only presented the monitoring findings obtained from the BBMEX campaign. However, the discussion on the what influences the thermal environment is limited. I suggest authors add some analysis in relation to thermal environment and factors (for instance, the linkage between surface temperature and air temperature), the influence of water ways/grass land on the air temperature, the influence of wind at different locations that the low-resolution monitoring system cannot meet. This point is also the innovative point of this paper.
Minor:
Line 42-43, authors have not respected the reduction of urban ventilation. Search several papers on the precinct ventilation zones and that verifying its potential for local warming mitigation.
Line 49-51, you may follow Oke's definition of local climate zone, change 'high-density' to compact.
Line 54, see papers on precinct ventilation zone that considers the relationships between urban morphology and urban ventilation.
Line 58, in urban climate studies, there are three levels, including meso-, local and micro. You may change the 'meter-scale' as high-resolution.
Line 60, since 'the' 1970s.
Line 132-145, not required for this paper. You may list the resolution of sensors in Table 2.
Author Response
Response to reviewer #3 comments
Authors gratefully thank to reviewer #3 for his/her thorough reviews and valuable comments which would contribute to improve the manuscript. Authors have revised the manuscript substantially to respond the reviewers’ comments. The reviewers’ all comments are responded and taken into account in the revised manuscript. In addition, many grammatical errors are corrected and incoherent sentences are rewritten throughout the manuscript. Reviewers’ comments are marked in black and the authors’ corresponding responses are marked in blue in the response. Major changes are marked in red in the revised manuscript. I hope that this manuscript will be accepted for the publication in Atmosphere.
Abstract, the current version of abstract is not comprehensive to inform readers with sufficient information. It should be rewritten to follow the background, research gap, research aim and objectives, the field work, findings and significance. The information in line 25-28 is not required.
- Abstract is rewritten to follow the background, objectives, field campaign, and finding. The information on L25-28 in the previous manuscript is simplified in the revised manuscript.
Introduction, the current version does not make sense to me, especially the research aim presented in the last paragraph is not to support scientific research but a practical issue. Therefore, authors are suggested restructuring the introduction to evidence a scientific question. You may start from line 55, to tell people that the existing monitoring system at the block, precinct, neighbourhood level is not supported by the high-resolution network and what is the limitation of them? And then you can tell people the significance of high-resolution monitoring network. However, the existing networks (Table 1) cannot necessarily support the local climate studies. "And you aim is to develop a local-scale high-resolution monitoring system and to test its applicability. Moreover, the final purposes should be understanding the local thermal environment.“
- The research questions to be answered are added on L76-80 in the revised manuscript. For the purpose of advancing to a meter-scale meteorology, more observation datasets are still needed. These could be used to quantify the air temperature difference between at several points in a building block and at the nearest synoptic weather station from the BB, to quantify the effects of materials and structures of BB on surface and air temperature in the BB, and to quantify the effects of facilities mitigating heat stresses on the temperatures in the BB.
Figure 4, you may add the wind information, as both surface and air temperatures are sensitive to areodynamics properties.
- Figure 4 is improved to be able to see wind vector clearly.
The results have only presented the monitoring findings obtained from the BBMEX campaign. However, the discussion on the what influences the thermal environment is limited. I suggest authors add some analysis in relation to thermal environment and factors (for instance, the linkage between surface temperature and air temperature), the influence of water ways/grass land on the air temperature, the influence of wind at different locations that the low-resolution monitoring system cannot meet. This point is also the innovative point of this paper.
- Discussion section is improved by adding the limitation of current state experiment (L428-433), applicability to LES model improvement (L440-442), necessity of more experiments in various BBs (L423-427), and applicability to urban planning and renewal for sustainable and resilient cities (L443-447).
- The detailed analysis on the temporal and spatial distribution of surface and air temperatures are now comparing with those computed by physical models.
Minor:
Line 42-43, authors have not respected the reduction of urban ventilation. Search several papers on the precinct ventilation zones and that verifying its potential for local warming mitigation.
- A recent datasets on the precinct ventilation performance conducted in Sydney, Australia, are referred on L73-74 in introduction (He et al., 2019).
Line 49-51, you may follow Oke's definition of local climate zone, change 'high-density' to compact.
- Following the terminology in Stewart and Oke (2012), the expression of high-density and low-density are changed to compact and open, respectively on L50-51 in the revised manuscript.
Line 54, see papers on precinct ventilation zone that considers the relationships between urban morphology and urban ventilation.
- A recent datasets on the precinct ventilation performance conducted in Sydney, Australia, are referred on L73-74 in introduction (He et al., 2019).
Line 58, in urban climate studies, there are three levels, including meso-, local and micro. You may change the 'meter-scale' as high-resolution.
- According to reviewer’s suggestion, ‘meter-scale’ is changed to ‘higher-resolution’. As for me, high-resolution could have too diverse and too ambiguous. High-resolution can have wide ranges from several hundred kilometer for global models to millimeter for direct numerical simulation (DNS).
Line 60, since 'the' 1970s.
è The word ‘the’ is added in front of 1970s on L59 in the revised manuscript.
Line 132-145, not required for this paper. You may list the resolution of sensors in Table 2.
- Net shortwave radiation, net longwave radiation, net radiation, and albedo were calculated using the observed 4-componets radiation. The friction velocity, friction temperature, and sensible heat flux are calculated using the 3-dimensional wind speed and sonic temperature sampled at 10 Hz.
- Accuracies of temperature sensors are added in Table 2.

Round 2
Reviewer 1 Report
Compared to the first version, the manuscript has become easier to read. I have still some comments which are mostly minor. Therefore, I recommend this manuscript for publishing in Atmosphere after minor revisions.
Remarks:
A space between the value and its unit is missing in some instances.
Detailed remarks:
l. 31: more energy than what?
l. 32: "in a day" - do you mean at daytime? And what about night-time?
l. 40-41: the first sentence still needs some work. For example you may write: "The urban population exceeded 4.2 billion in 2019 and is projected to reach 6.3 billion by 2050 (x% of global population)" - exchange x by the actual value.
l. 66: I would rather write "High-rise building blocks" instead of "High-rise buildings"
l. 68: change "micro-scale observation in an urban areas" to "micro-scale observation projects in urban areas"
l. 73: change "Recently, the 52" to "Recently, 52"
l. 77: change "between at several several points in a BB and at the" to "between several points in a BB and to the"
l.113: what does "12.23 Fountain" mean?
l. 114: change "are installed at" to "are located at"
Table 2: in the text and in Figure 3a you refer to the AWS as "Type A" and "Type B". Please correct in the table.
l. 268: change "18 were selected points in" to "18 locations were selected in"
l. 278: horizontal and vertical distributions
ll. 282-283: "low surface temperature area" - this is only seen at 8:30 LST correct?
l. 292-293: you might just write that this depends on the solar incidence angle.
l. 293-293: I think "faces" needs to be "facades" here
Fig. 8b: the addition of air temperature of a nearby AWS would be informative for the reader to infer sensible heat fluxes from the surfaces. E.g. I'd like to see how close the tree temperatures were to the air temperatures.
l. 325-326: this belongs in the Discussion section
l. 339-341: I do not understand this sentence. Maximum normally implies that 1 value is given, not 2 values.
l. 346-347: "could have been trapped inside the BB" - I do not see what the physical mechanism for this could be. Under unstable atmospheric conditions the warmest air should be in the lowest layers. So I do not see any other possible explanation than anthropogenic heat being released near that height. Besides that, the reasons for this should be discussed in the Discussion section not here.
l. 363: "indicated" can be changed to "show" or similar.
l. 377-378: I do not understand this sentence and I guess it belongs again in the Discussion section and needs further explanation.
l. 379: change "LST was mainly" to "LST mainly"
Fig. 11b: if you map this data eventually with spatial interpolation the reader could instantly see what you describe in the text.
l. 399: "Observer" - do you mean "volunteering observers"?
l. 414/415: change "in a BB" to "in the BB"
l.433 : you should mention that this would require additional measurements with sensors being attached to these surfaces which eventually was not feasible in your study.
l. 441-442: what do you mean by "more sophisticated physical processes should be developed"? Do you mean more elaborate measurements? Please clarify.
Author Response
Response to Reviewer’s comments
Authors thanks to reviewer #1 for his/her valuable and detailed comments, which will improve this manuscript substantially. All comments are answered in the revised manuscript. The reviewer’s comments are listed in Black, and corresponding responses are marked in Blue. Major changes are marked in Red in the revised manuscript. I hope that this manuscript will be accepted for the publication in the Atmosphere.
A space between the value and its unit is missing in some instances.
- Authors corrected missing spaces between the value and its unit throughout the manuscript.
Detailed remarks:
- 31: more energy than what?
- ‘than over flat surfaces’ is added after ‘in BBs’ on L31 in the revised manuscript.
- 32: "in a day" - do you mean at daytime? And what about night-time?
- The phrase is rewritten as ‘by 0.1−2 °C (1.1−1.9 °C) in daytime (nighttime)’ on L32-33 in the revised manuscript.
- 40-41: the first sentence still needs some work. For example you may write: "The urban population exceeded 4.2 billion in 2019 and is projected to reach 6.3 billion by 2050 (x% of global population)" - exchange x by the actual value.
- According the reviewers’ suggestion, the phrase is rewritten as ‘exceeded 4.2 billion in 2019 and is projected to reach 6.3 billion (65 % of world population) by 2050’ on L40-41 in the revised manuscript.
- 66: I would rather write "High-rise building blocks" instead of "High-rise buildings"
- ‘High-rise buildings’ is changed to ‘High-rise building blocks’ on L65 in the revised manuscript.
- 68: change "micro-scale observation in an urban areas" to "micro-scale observation projects in urban areas"
- The phrase ‘micro-scale observation in an urban areas’ is changed to ‘micro-scale observation projects in urban areas’ on L67 in the revised manuscript.
- 73: change "Recently, the 52" to "Recently, 52"
- ‘Recently, the 52’ is changed to ‘Recently, 52’ on L72 in the revised manuscript.
- 77: change "between at several several points in a BB and at the" to "between several points in a BB and to the"
- ‘between at several points in a BB and at the’ is changed to ‘between several points in a BB and to the’ on L76-77 in the revised manuscript.
l.113: what does "12.23 Fountain" mean?
- ’12.23 Fountain’ is the name of the ground fountain. ‘the’ is added before 12.23 fountain on L113 in the revised manuscript.
- 114: change "are installed at" to "are located at"
- The phrase ‘are installed at’ is changed to ‘are located at’ on L114 in the revised manuscript.
Table 2: in the text and in Figure 3a you refer to the AWS as "Type A" and "Type B". Please correct in the table.
- ‘Type I’ and ‘Type II’ are changed to ‘Type A’ and ‘Type B’ in Table 2, respectively.
- 268: change "18 were selected points in" to "18 locations were selected in"
- The phrase ’18 were selected points in’ is changed to ’18 locations were selected in’ on L268 in the revised manuscript.
- 278: horizontal and vertical distributions
- ‘horizontal distribution’ is changed to ‘horizontal and vertical distribution’ on L278 in the revised manuscript.
- 282-283: "low surface temperature area" - this is only seen at 8:30 LST correct?
- The low surface temperature area was shown at 0750 LST and 0830 LST on L283-284 in the revised manuscript.
- 292-293: you might just write that this depends on the solar incidence angle.
- The sentence is rewritten as ‘The temporal variation in the surface temperature directly depends on the solar zenith or elevation angles at a given surface, which is a function of slopes of facades, and shade/sunlit by surrounding obstacles’ on L293-294 in the revised manuscript.
- 293-293: I think "faces" needs to be "facades" here
- ‘faces’ is changed to ‘facades’ on L294 in the revised manuscript.
Fig. 8b: the addition of air temperature of a nearby AWS would be informative for the reader to infer sensible heat fluxes from the surfaces. E.g. I'd like to see how close the tree temperatures were to the air temperatures.
- Time series of the Seoul Station is added in Figure 8b.
- 325-326: this belongs in the Discussion section
- The sentence is removed in the revised manuscript.
- 339-341: I do not understand this sentence. Maximum normally implies that 1 value is given, not 2 values.
- The sentence is rewritten as the followings in the revised manuscript: Maximum difference of air temperature between A1 and A2 was 0.9 °C at 1042 LST, while that between A4 and A3 was 2.6 °C at 1535 LST.
- 346-347: "could have been trapped inside the BB" - I do not see what the physical mechanism for this could be. Under unstable atmospheric conditions the warmest air should be in the lowest layers. So I do not see any other possible explanation than anthropogenic heat being released near that height. Besides that, the reasons for this should be discussed in the Discussion section not here.
- The phrase ‘could have been trapped inside the BB in a daytime’ is removed on L346 in the revised manuscript.
- 363: "indicated" can be changed to "show" or similar.
- ‘indcated’ is changed to ‘show’ on L362 in the revised manuscript.
- 377-378: I do not understand this sentence and I guess it belongs again in the Discussion section and needs further explanation.
- The sentence is moved to discussion Section (L439-443) in the revised manuscript.
- 379: change "LST was mainly" to "LST mainly"
- ‘LST was mainly’ is changed to ‘LST mainly’ on L376 in the revised manuscript.
Fig. 11b: if you map this data eventually with spatial interpolation the reader could instantly see what you describe in the text.
- Horizontal distribution of surface and 1.5 m air temperature during 1600-1640 LST are added in Figure 11c and d in the revised manuscript.
- 399: "Observer" - do you mean "volunteering observers"?
- ‘Observer’ is a meteorological instrument company. So ‘Observer’ is changed to ‘Observer Inc.’ on L86 and L399 in the revised manuscript.
- 414/415: change "in a BB" to "in the BB"
- The term ‘in a BB’ is changed to ‘in the BB’ on L413 in the revised manuscript.
l.433 : you should mention that this would require additional measurements with sensors being attached to these surfaces which eventually was not feasible in your study.
- The last sentence is rewritten as the followings on L431-432 in the revised manuscript: The exact emissivity for each pixel could be determined by comparing the contact-type and the infrared-type surface temperatures.
- 441-442: what do you mean by "more sophisticated physical processes should be developed"? Do you mean more elaborate measurements? Please clarify.
The sentence is rewritten as ‘more sophisticated energy transfer process between wall of a building and air inside the BB should be parameterized on the basis of observation and verified’ on L448-450 in the revised manuscript.

Reviewer 2 Report
I thank the authors for their further work on the manuscript and considering my remarks.
As stated in the first review round I find the study very valuable and up to date.
However, in my opinion the discussion is still very scarce, there is actually mainly a summary and a substantial scientific discussion is missing. I recommend the authors to add a section on how the findings fit with existing literature. Are the results in line with other findings or contrary? Give some examples from other studies, discuss them.
Author Response
Authors thanks to reviewer #2 for his/her valuable and detailed comments, which will improve this manuscript substantially.
According to the reviewer’s suggestion, authors separate Section 4 into 2 subsections of ‘summary’ and ‘discussions’. The discussion subsection (Section 4.2, L419-461) was much improved by including necessity of more datasets, weakness in data process (TIR and eddy covariance), affecting factors on surface and air temperatures, application to model improvement and further research topics, application to smart cities. Detailed parameterization on energy transfer between a given vertical or sloped surface and the surrounding air are now on hands.
Major changes are marked in Red in the revised manuscript. I hope that this manuscript will be accepted for the publication in the Atmosphere.

Reviewer 3 Report
Well done
Author Response
Authors thanks to reviewer #3 for his/her valuable and detailed comments, which will improve this manuscript substantially. I hope that this manuscript will be accepted for the publication in the Atmosphere.